# Development of a CRISPR activation system for targeted gene upregulation in *Synechocystis* sp. PCC 6803
Barbara Bourgade [1,2], Hao Xie[1,3], Peter Lindblad [1] & Karin Stensjö [1] ✉

The photosynthetic cyanobacterium *Synechocystis* sp. PCC 6803 offers a promising sustainable solution for simultaneous $CO_2$ fixation and compound bioproduction. While various heterologous products have now been synthesised in *Synechocystis*, limited genetic tools hinder further strain engineering for efficient production. Here, we present a versatile CRISPR activation (CRISPRa) system for *Synechocystis*, enabling robust multiplexed activation of both heterologous and endogenous targets. Following tool characterisation, we applied CRISPRa to explore targets influencing biofuel production, specifically isobutanol (IB) and 3-methyl-1-butanol (3M1B), demonstrating a proof-of-concept approach to identify key reactions constraining compound biosynthesis. Notably, individual upregulation of target genes, such as *pyk1*, resulted in up to 4-fold increase in IB/3M1B formation while synergetic effects from multiplexed targeting further enhanced compound production, highlighting the value of this tool for rapid metabolic mapping. Interestingly, activation efficacy did not consistently predict increases in compound formation, suggesting complex regulatory interactions influencing bioproduction. This work establishes a CRISPRa system for targeted upregulation in cyanobacteria, providing an adaptable platform for high-throughput screening, metabolic pathway optimisation and functional genomics. Our CRISPRa system provides a crucial advance in the genetic toolbox available for *Synechocystis* and will facilitate innovative applications in both fundamental research and metabolic engineering in cyanobacteria.

Biological processes are increasingly recognised as robust, sustainable alternatives to conventional chemical methods for the production of fuels and chemicals. Amongst these, photosynthetic cyanobacteria offer a promising strategy for simultaneous $CO_2$ fixation and compound bioproduction from sunlight, a critical approach to mitigate the escalating climate crisis. The model freshwater cyanobacterium *Synechocystis* sp. PCC 6803 (hereafter referred to as *Synechocystis*) has become a leading candidate in photosynthetic biotechnology, due to its well-characterised physiology and metabolism. Through the introduction of heterologous pathways and redirection of metabolic fluxes, *Synechocystis* has demonstrated its potential as a versatile microbial cell factory, enabling the laboratory-scale production of a wide range of compounds[1–7]. However, despite these advancements, engineering cyanobacteria as efficient microbial cell factories remains a significant challenge. In particular, while *Synechocystis* benefits from a more developed genetic toolkit compared to other cyanobacteria, the available

tools still lack the diversity and versatility seen in model bacteria, restricting complex metabolic engineering and limiting efforts to optimise compound production.

In recent years, CRISPR-Cas (Clustered Regularly Interspaced Short Palindromic Repeats—CRISPR associated proteins) technologies have significantly expanded and accelerated the scope of genetic engineering[8], enabling precise gene-editing and transcriptional control. Particularly noteworthy are CRISPR interference (CRISPRi)[9] and CRISPR activation (CRISPRa)[10,11], which utilise catalytically deactivated Cas (dCas) enzymes to precisely regulate transcription. CRISPRi represses transcription by blocking transcription initiation or elongation, while CRISPRa enhances transcription by recruiting RNA polymerase at a target locus. In prokaryotic systems, CRISPRa-mediated gene activation has been achieved using various mediators, such as the *Escherichia coli* transcriptional activator SoxS or RNA polymerase subunits[11,12], facilitated through synthetic protein fusions[13]

[1]Microbial Chemistry, Department of Chemistry-Ångström Laboratory, Uppsala University, Uppsala, Sweden. [2]Present address: Department of Organismal Biology, Uppsala University, Uppsala, Sweden. [3]Present address: College of Bioengineering, Sichuan University of Science and Engineering, Yibin, China. ✉ e-mail: karin.stensjo@kemi.uu.se

or RNA-binding proteins, such as the bacteriophage MS2-MCP (MS2 coat protein) system[11,14]. These transient and reversible techniques allow systematic exploration of gene function and regulation as well as comprehensive metabolic studies to identify key reactions that limit compound biosynthesis. For example, CRISPRi libraries have been employed for genome-wide screens to assess gene essentiality under diverse growth conditions in *Synechocystis*, revealing essential metabolic processes required for survival in specific environments[15,16]. Targeted CRISPRi approaches have also been utilised to investigate photosynthetic complexes[17] and identify metabolic pathways constraining biosynthesis of target compounds[18]. Collectively, these studies highlight the broad applicability and significance of CRISPR-mediated regulatory tools. However, in contrast to CRISPRi, CRISPRa remains largely unexplored in cyanobacteria, despite its clear potential to facilitate rapid metabolic mapping and target discovery in both fundamental and applied research. A recent study[19] demonstrated targeted gene activation in *Synechococcus elongatus* PCC 7942 by fusing the native RNA polymerase ω subunit (RpoZ) to dCas9, which was subsequently utilised to enhance isopentenol production. This highlights the potential of CRISPRa as a powerful tool for metabolic engineering in cyanobacteria, offering new opportunities to fine-tune gene expression and optimise biosynthetic pathways.

In this study, we describe the development of a rhamnose-inducible CRISPRa system for targeted gene upregulation in *Synechocystis* using a dCas12a-SoxS protein fusion (Fig. 1) to recruit RNA polymerase at specific promoters. Our system demonstrates reliable gene activation across various targets, with activation efficacy dependent on guide RNA (gRNA) position relative to the transcriptional start site (TSS) and the intrinsic strength of the target promoter. We subsequently aimed to evaluate its applicability for metabolic engineering by applying CRISPRa to upregulate key genes involved in the biosynthesis of the biofuel candidates isobutanol (IB) and 3-methyl-1-butanol (3M1B). This approach successfully identified specific genes that resulted in increased IB and 3M1B titres when upregulated, establishing a proof-of-concept application of CRISPRa for metabolic engineering in cyanobacteria, and further highlighted the complex and dynamic system-wide metabolic regulation influencing compound bioproduction. Our study introduces a novel cyanobacterial CRISPRa system that enables targeted transcriptional upregulation, an approach previously inaccessible with existing cyanobacterial genetic tools. This advancement significantly broadens the engineering possibilities for *Synechocystis* and establishes CRISPRa as a versatile and scalable platform to enhance cyanobacterial biotechnology, offering a powerful approach for optimising photosynthetic production of high-value compounds.

## Results

### Development and characterisation of a CRISPRa system for targeted gene upregulation in *Synechocystis*

To establish a CRISPRa system in *Synechocystis*, we fused the R93A variant[20] of the *E. coli* transcriptional activator SoxS to the C-terminus of *Francisella novicida* catalytically inactive dCas12a mutant with a 10-aa linker peptide[13] (Fig. 1). This design was informed by two considerations: first, SoxS[R93A] has been shown to enhance CRISPR activity relative to wild-type SoxS in other organisms[19], and second, the dCas12 variant has been successfully applied for CRISPRi in *Synechocystis*[17,18]. The resulting fusion construct was placed under the control of the rhamnose-inducible $P_{rha}$ promoter[21] (Supplementary Fig. 1), allowing precise induction of gene activation.

Our first goal was to achieve functional CRISPR-mediated activation in *Synechocystis* and further characterise the behaviour of this system using a fluorescence-based approach. Prior studies of prokaryotic dCas9-based CRISPRa systems have demonstrated that the activation efficacy is highly sensitive to the position of the gRNA target site relative to the TSS of the candidate gene[20,22–25]. To assess whether this positional effect influences our dCas12a-based CRISPRa system, we explored activation of a chromosomally integrated green fluorescent protein (GFP) gene driven by the strong constitutive $P_{trc}$ promoter[26] using different gRNAs positioned across a 328-bp region upstream $P_{trc}$ (Fig. 2). Our results showed clear GFP upregulation upon rhamnose induction with gRNAs targeting sites between −97 and −328 compared to the negative control (EV) lacking a targeting gRNA, indicating a flexible targeting range. Notably, gRNAs positioned between −97 and −156 produced the highest fluorescence levels, suggesting that this region represents the optimal targeting window for gene activation with dCas12a-SoxS in *Synechocystis*, by effectively recruiting RNA polymerase to the target promoter. In contrast, a gRNA positioned at −48 failed to activate GFP, possibly due to disrupted interactions between SoxS and RNA polymerase. Additionally, previous studies have reported strand-specific differences in activation levels when targeting the DNA template strand versus non-template strand[24]. We further explored this behaviour using two overlapping pairs of gRNAs (−97 NTS/−108 and −144 NTS/−156) targeting the template or non-template strand within the optimal target window. Our results indicated that targeting the non-template strand led to enhanced fold-activation, providing an additional strategy for fine-tuning activation levels. Consistent with previous reports on dCas12-based systems[27], co-expression of two gRNAs (−108/−156) demonstrated a combinatorial effect, resulting in fold-activation levels exceeding those of the respective single gRNAs. Additionally, the CRISPRa system was also applicable for transient gene repression by directing it to *gfp* coding sequence (CDS), effectively blocking transcription elongation, as previously described for similar systems[13]. This dual capability for upregulation and downregulation provides the opportunity for complex metabolic rewiring through simultaneous targeting of multiple genes.

To assess the stability of this CRISPRa system, we monitored GFP fluorescence over a 96-hour post-induction period using four selected gRNAs (−48; −108; −156, and CDS). GFP upregulation was maintained throughout this period (Supplementary Fig. 2), highlighting the system's applicability for target mapping over multiple days. Moreover, based on prior results showing effective CRISPRi repression with 3 mM of rhamnose[18], our initial characterisation was conducted with 3 mM of rhamnose. We subsequently explored whether higher inducer concentrations correlated with increased activation levels by testing the effect of 6 or 9 mM of rhamnose on GFP activation with selected gRNAs (−97 NTS; −108; −144 NTS, and −156). These results indicated that increased rhamnose concentrations did not correspond to higher fold-activation (Supplementary Fig. 3), suggesting a saturation point for inducer efficacy. Based on this, we conducted subsequent experiments using 3 mM of rhamnose for CRISPRa induction.

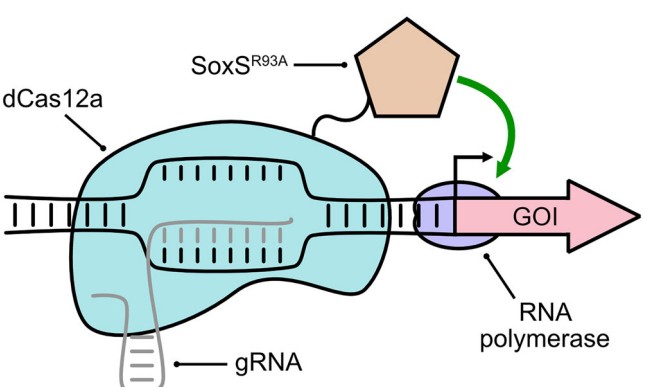

**Fig. 1 | The developed CRISPRa system consists of a protein fusion between *Francisella novicida* dCas12a and SoxS[R93A].** Through gRNA-guided dCas12a binding at a specific site upstream of the target promoter, RNA polymerase is recruited by SoxS to initiate transcription at the candidate promoter. GOI: gene of interest.

**Fig. 2 | GFP fluorescence at 72 hours post-induction with 3 mM rhamnose with eight gRNAs targeting $P_{trc}$ with dCas12a-SoxS. a** Seven gRNAs were designed to target a 328-bp region upstream $P_{trc}$ driving GFP expression, with an additional gRNA targeting the coding sequence. **b** GFP fluorescence was measured in the absence or presence of the rhamnose inducer across 8 candidate gRNAs. **c** Fold-activation of GFP was determined relative to the negative control (EV) for each candidate gRNA. GFP green fluorescent protein, NTS non-template strand, EV negative control—sBB_CA1 expressing plasmid pBB_CA, which contains dCas12a-SoxS and a CRISPR array lacking a targeting gRNA, CDS coding sequence. Error bars indicate standard deviation (− rhamnose samples: EV: $n = 6$; −48, −108, −156, CDS: $n = 10$; −97 NTS, −144 NTS, −251, −328, −108/−156: $n = 3$; + rhamnose samples: EV: $n = 8$; −97 NTS: $n = 5$; other gRNAs: $n = 10$). $p$ value representation: ns > 0.05; * < 0.05; ** < 0.01; *** < 0.001. $p$ value was calculated by comparing each sample to the negative control unless otherwise stated.

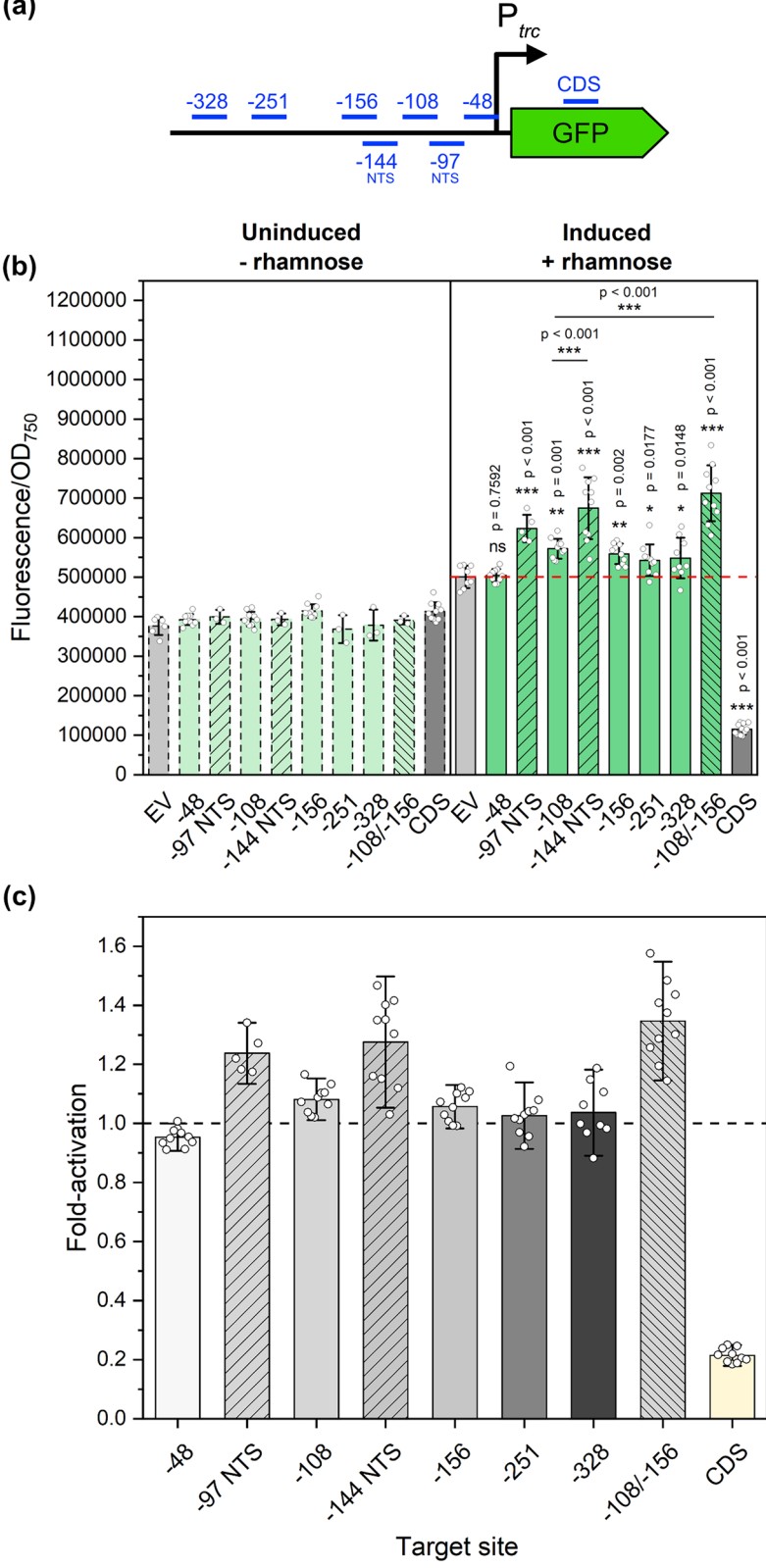

## Promoter strength-dependent activation levels using the CRISPRa dCas12a-SoxS system

Previous prokaryotic CRISPRa systems have demonstrated that activation levels are closely linked to the intrinsic strength of the target promoter[20,24], with minimal response observed for strong promoters and high fold-activation achieved with weaker promoters. To investigate whether this

effect occurs with our CRISPRa system, we selected four synthetic promoters from the modified BioBrick_J231XX series, previously characterised in *Synechocystis*[28]. These promoters differ by specific point mutations within the promoter sequence, allowing us to use the same gRNA to target upstream all promoters and thereby isolate the effects of promoter strength from those of gRNA positioning and sequence. Consistent with previous

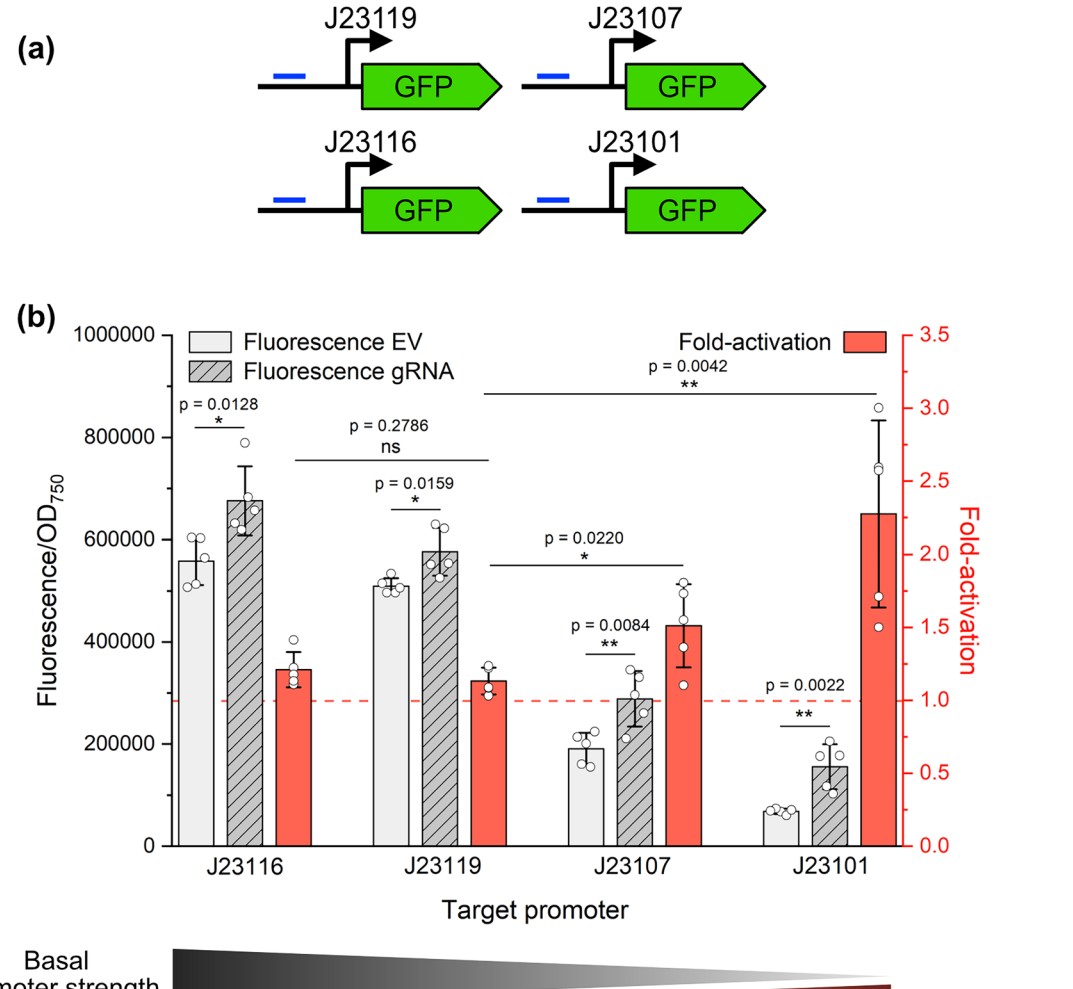

**Fig. 3 | Correlation between promoter strength and activation levels with the dCas12a-SoxS CRISPRa system. a** Four synthetic promoters of the Biobrick J23 series were selected and tested for activation with dCas12a-SoxS using a common gRNA. **b** Fluorescence (grey) and activation (red) levels were measured following rhamnose induction of the dCas12a-SoxS system for the four J23 promoters. EV negative control—respective background strains expressing plasmid pBB_CA. Error bars indicate standard deviation ($n = 5$). $p$ value representation: ns > 0.05; * < 0.05; ** < 0.01; *** < 0.001.

reports[20,24], our results indicated that activation was indeed stronger for weaker promoters (Fig. 3). Notably, we achieved a 2.28 (±0.57) fold-activation with the weakest promoter tested, J23101, compared to 1.21 (±0.10) with the strong promoter J23116. These findings suggest that basal promoter strength is a key factor in determining fold-activation in response to targeting with our CRISPRa system, which may constrain upregulation of endogenous genes.

## CRISPRa-mediated upregulation of *kivD*^S286T^ for enhanced IB and 3M1B biosynthesis in *Synechocystis*

We then aimed to evaluate the applicability of our CRISPRa system for metabolic engineering in *Synechocystis*. We thus investigated CRISPRa-mediated upregulation of the engineered *kivD*^S286T^ gene[29], which encodes an α-ketoisovalerate decarboxylase, enabling the biosynthesis of IB and 3M1B from 2-ketoisovalerate and 2-ketoisocaproate, respectively, in *Synechocystis*. Our experiments included three background strains: ddh_kivD (obtained with plasmid pHX8[30]; single *kivD*^S286T^ copy inserted at the *ddh* site), HX11 (as detailed in Xie et al.[18]; two *kivD*^S286T^ copies inserted at the *ddh* and NS1[4] *(slr0168)* sites) and HX51 (three *kivD*^S286T^ copies inserted at the *ddh*, NS1 and *sll1564*[4] sites). These strains contained one, two, or three chromosomally integrated copies of *kivD*^S286T^ under the control of the P*trc* promoter (Fig. 4) and were subjected to single, dual, and triple CRISPRa targeting. Based on

our previous findings identifying an optimal targeting window for CRISPRa, we designed gRNAs binding within this window for each target locus (*ddh*: −115; NS1: −156; *sll1564*: −173) to maximise gene activation.

IB and 3M1B production was monitored over 10 days, with day 4 consistently showing the highest CRISPRa-induced improvement across all strains (Fig. 4; Supplementary Fig. 4). At later timepoints, IB and 3M1B levels became comparable between CRISPRa-targeted and non-targeted strains. Notably, only modest improvements in product titres were achieved with CRISPRa-mediated *kivD*^S286T^ upregulation across most gRNA combinations. In HX51-derived strains, only dual and triple targeting showed slight improvements in IB and 3M1B production, suggesting that single targeting was insufficient to enhance product titres in this background strain. These findings indicate that elevated expression of the key biosynthetic gene may not directly translate to higher product titres, potentially due to substrate limitations acting as a bottleneck for product formation.

Additionally, gRNA performance varied depending on the strain background, as demonstrated by the opposite effects of dual NS1 and *ddh* targeting in HX11 and HX51. In HX11, co-targeting these two loci led to a modest increase of IB and 3M1B production, whereas, in contrast, the same approach in HX51 unexpectedly reduced IB and 3M1B titres. Since identical *kivD*^S286T^ constructs and gRNAs were used for gene activation, this observation indicates that the genetic context of the respective strains, rather than

**Fig. 4 | Relative isobutanol (IB) and 3-methyl-1-butanol (3M1B) titres on days 4 and 8 in response to CRISPRa-mediated kivD^S286T upregulation.** Single, dual, and triple CRISPRa targeting using different gRNA combinations were assessed in three background strains: **a** *ddh_kivD* (harbouring one copy of *kivD^S286T*), **b** HX11 (two copies of *kivD^S286T*), and **c** HX51 (three copies of *kivD^S286T*), as shown in the corresponding construct schematics. The gRNA(s) expressed in each resulting strain are represented on the×axis. For dual targeting, both gRNAs are named. Triple targeting of ddh, NS1, and sll1564 loci in HX51 is referred to as "Triple". Relative titres were calculated in comparison to the respective negative control (EV) for each background strain. EV negative control—respective background strain expressing plasmid pBB_CA, UP upstream homology arm, DN downstream homology arm. Error bars indicate standard deviation ($n = 3$). *p* value representation: * < 0.05; ** < 0.01.

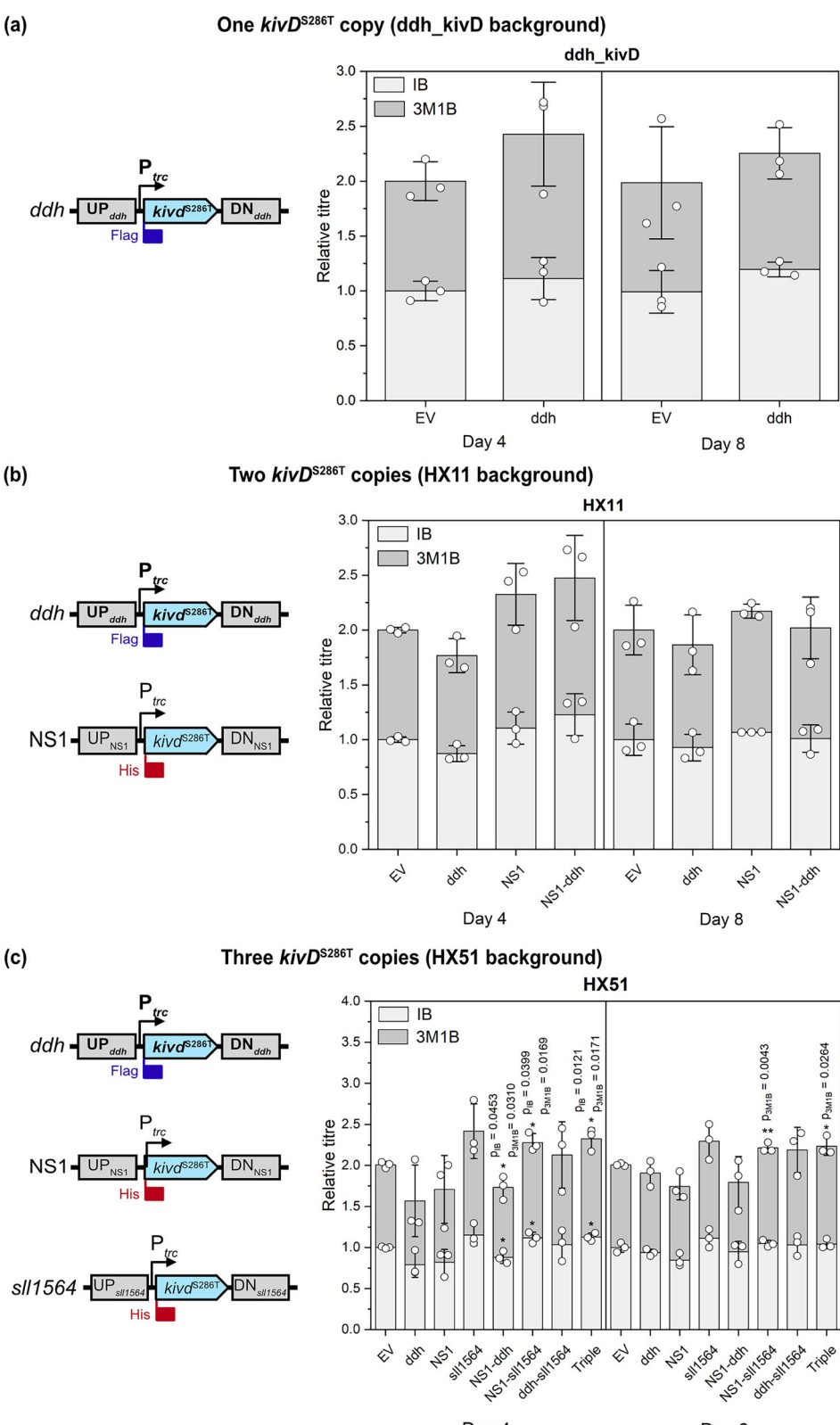

gRNA efficiency, may account for these differences. Unexpectedly, CRISPRa targeting led to a modest reduction in IB and 3M1B production in several strains, particularly prevalent in HX51 derivatives. Analysis of relative *kivD^S286T* transcript levels on day 4 (Fig. 5) confirmed that CRISPRa-mediated upregulation was effective, although this did not consistently correlate with proportional improvements in product titres. For example,

despite a 3.19 (±0.24) fold-activation in the targeted ddh_kivD strain, only marginal improvements in IB and 3M1B formation were observed. In HX11-derived strains, lower activation levels and delayed growth (Supplementary Fig. 4) suggest that the reduced response may be linked to an unidentified phenotype in the HX11 background strain. In HX51 derivatives, fold-activation generally correlated with product titres, confirming

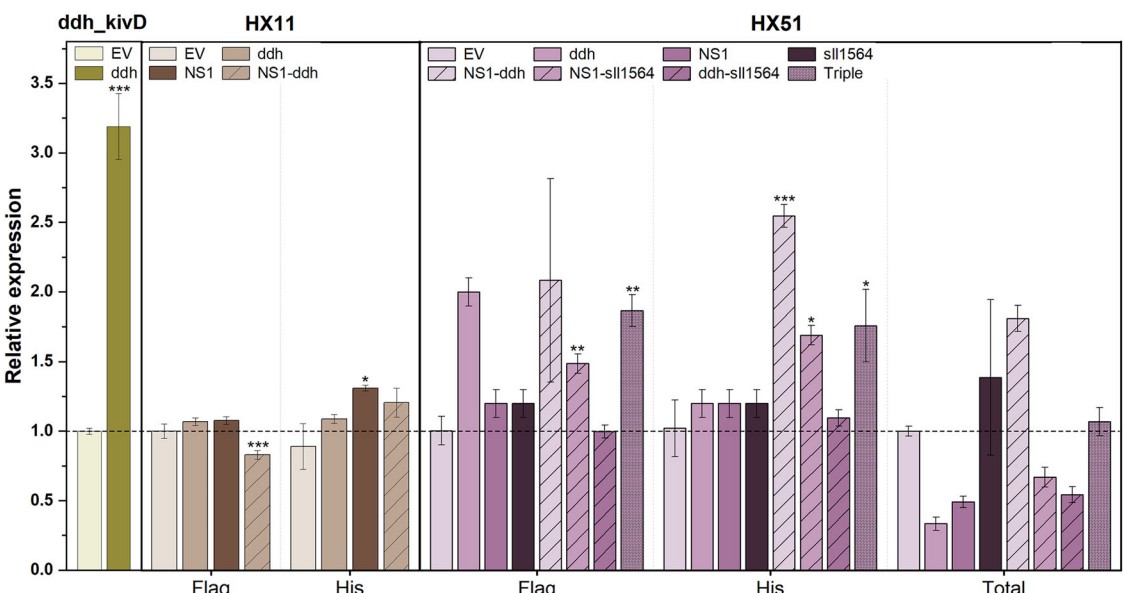

**Fig. 5 | Relative $kivD^{S286T}$ transcript levels at day 4 in CRISPR-activated strains.** Gene expression was quantified with different primer sets designed to bind within the coding sequence (Total), the Flag tag (Flag) or the His tag (His) to differentiate which $kivD^{S286T}$ copy was activated when applicable. EV negative control—respective background strain expressing plasmid pBB_CA. Error bars indicate standard deviation ($n = 3$). $p$ value representation: * < 0.05; ** < 0.01; *** < 0.001.

that single targeting unexpectedly led to $kivD^{S286T}$ downregulation. However, transcript levels of $kivD^{S286T}$ at the NS1 and $sll1564$ sites could not be differentiated in qPCR readings, limiting our ability to fully characterise the tool's behaviour in these strains. Collectively, these findings highlight the complexity of CRISPRa-mediated metabolic engineering in *Synechocystis*, suggesting that factors beyond transcriptional upregulation, such as substrate availability and strain-specific genetic context, play critical roles in determining biosynthetic output.

## CRISPRa-mediated metabolic mapping to identify candidate genes enhancing IB and 3M1B biosynthesis

To further test the applicability of our system for metabolic engineering, we aimed to investigate CRISPR-mediated activation of several endogenous genes. We thus applied our system to upregulate candidate genes (Fig. 6) hypothesised to influence pyruvate formation and cofactor availability, both potentially limiting IB/3M1B biosynthesis. We upregulated $pyk1$ ($sll0587$) and $pyk2$ ($sll1275$)[31], encoding pyruvate kinase 1 and 2, respectively, to increase the conversion of phosphoenolpyruvate (PEP) to pyruvate. Additionally, we explored enhancing pyruvate availability by targeting $me$ encoding a malic enzyme[32], which catalyses the oxidation of malate to pyruvate, as previous studies have demonstrated that $me$ overexpression leads to increased formation of pyruvate-derived products, such as lactate[33]. To further channel carbon flow from $CO_2$ fixation towards pyruvate formation, we targeted triosephosphate isomerase (TPI), which converts dihydroxyacetone phosphate (DHAP) into glyceraldehyde-3-phosphate (G3P) in the Calvin-Benson-Bassham (CBB) cycle. This approach has previously increased phloroglucinol biosynthesis in *E. coli*[34] while overexpression of *E. coli* TPI, in conjunction with other enzymes, has improved IB/3M1B formation[4] in *Synechocystis*. In addition to genes involved in pyruvate formation, we also targeted two additional genes relevant for cofactor availability, specifically NADPH, hypothesising that limited availability of NADPH may constrain the conversion of 2-acetolactate to 2,3-dihydroxyisovalerate[35], as reported in other organisms[36], as well as the conversion of isobutyraldehyde and 3-methylbutyraldehyde into the respective products IB and 3M1B. In a previous study, overexpression of $pntAB$, coding for a membrane-bound pyridine nucleotide transhydrogenase, slightly increased L-valine biosynthesis in *E. coli*, by promoting cofactor regeneration[36]. We thus chose to target $pntA$ for upregulation,

although it has previously been reported that PntA is particularly important for cofactor regeneration under mixotrophic growth conditions, but is dispensable under phototrophic conditions[37] in *Synechocystis*. Lastly, we targeted $petH$, encoding a ferredoxin-NADP$^+$ oxidoreductase, a key enzyme for NADPH regeneration in the photosynthetic electron transport chain[38], which has been demonstrated to increase intracellular NADPH levels when overexpressed in *Synechocystis*[39]. For three selected genes ($pyk1$, $pyk2$, and $pntA$), we tested two different gRNAs, referred to as gene.1 and gene.2 on Fig. 7 and Supplementary Fig. 7, both targeting within the optimal targeting window between −100 to −150 bp upstream of the TSS, in order to investigate the impact of the gRNA sequence on activation levels.

Upon CRISPRa targeting, IB and 3M1B biosynthesis was increased for most targets (Fig. 7), although some targets displayed only transient effects. In particular, upregulation of genes hypothesised to affect pyruvate availability, as exemplified by gRNAs pyk2.2 and tpi, improved IB/3M1B primarily at early growth stages, with minimal effects on day 4 (Fig. 7a, b). In contrast, activation of $petH$ sustained increased product biosynthesis throughout the experimental period. Notably, across all targets, improvements were more pronounced for 3M1B than IB, reaching up to ~3.5-fold improvement on day 2 (Fig. 7c). Additionally, negligible differences in compound production were detected between the two candidate gRNAs for $pyk1$, $pyk2$ and $pntA$, suggesting minimal influence of gRNA sequence within the optimal targeting window. However, strains targeting $pyk1$ and $pyk2$ displayed distinct production profiles, highlighting their unique roles and regulation. Notably, strains with $pyk1$ activation exhibited high biological variability, associated with impaired growth (Supplementary Fig. 4), particularly noticeable with gRNA pyk1.2. These strains also displayed a significant shift in the IB/3M1B ratio (Fig. 7d), indicating substantial metabolic disruption. Although the IB/3M1B ratio was also affected across several targets on day 2 (Supplementary Fig. 6), $pyk1$ activation led to the most pronounced and lasting effect throughout the experiment, which requires further system-wide exploration to elucidate.

We confirmed successful upregulation of all candidate genes upon CRISPRa targeting, with strong activation observed across most targets (Supplementary Fig. 7). Minimal variation was detected between different gRNAs targeting the same gene, highlighting the efficacy of targeting within the optimal window to achieve robust activation, even for endogenous genes. However, this pattern did not extend to $pyk2$, suggesting that further

**Fig. 6 | Simplified biosynthetic pathway for iso-butanol and 3-methyl-1-butanol formation in _Synechocystis_.** The key heterologous enzyme KivD (red) catalyses the conversion of 2-ketoisovalerate and 2-ketoisocaproate into isobutylaldehyde and 3-methylbutyraldehyde, respectively, which are subsequently converted into isobutanol and 3-methyl-1-butanol by _Synechocystis_ alcohol dehydrogenase (Adh). Genes (blue) targeted by the CRISPRa system were selected to enhance pyruvate formation and NADPH regeneration, hypothesised as key bottlenecks for IB/3M1B biosynthesis. Adh alcohol dehydrogenase, CBB cycle Calvin-Benson-Bassham cycle, DHAP dihydroxyacetone phosphate, FNR ferredoxin-NADP$^+$ oxidoreductase, G3P glyceraldehyde-3-phosphate, KivD α-ketoisovalerate decarboxylase, ME malic enzyme, PEP phosphoenolpyruvate, PK pyruvate kinase, PntAB pyridine nucleotide transhydrogenase, PSI photosystem I, TCA cycle tricarboxylic acid cycle.

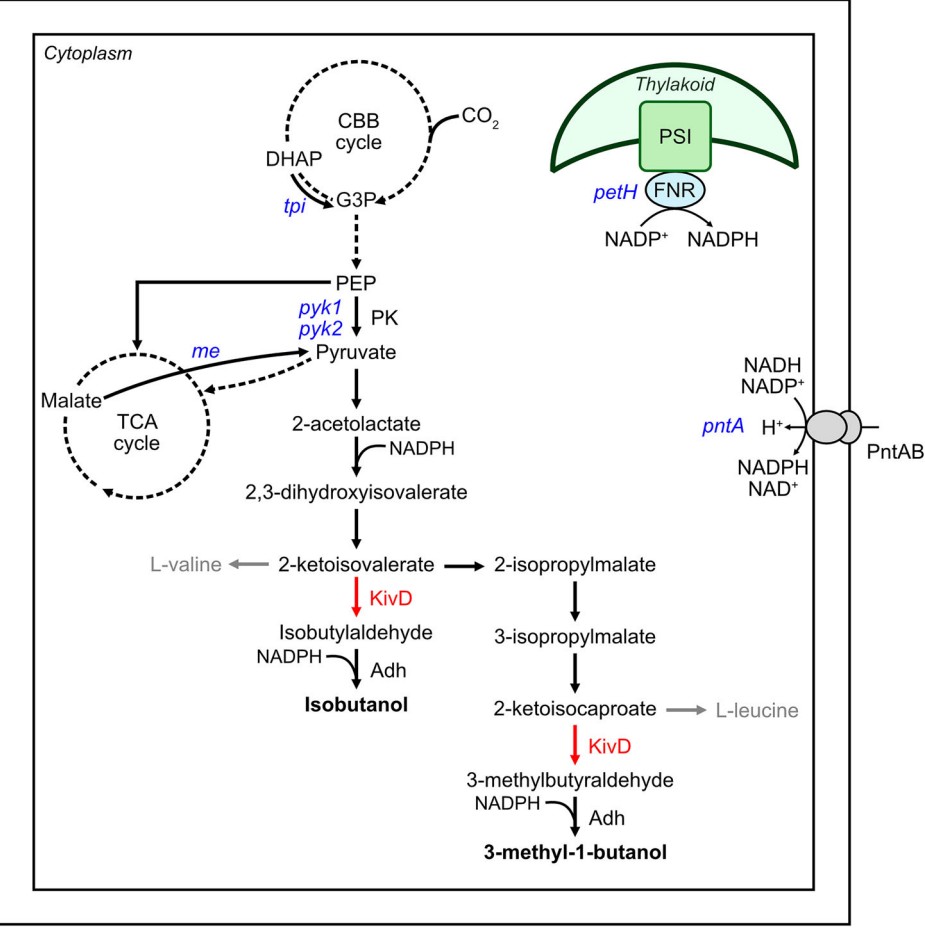

gRNA optimisation may be necessary for specific targets. Importantly, we found no direct correlation between activation efficiency and production increase, as demonstrated by the moderate improvements in product formation when activating _petH_, despite its strong activation. These results highlight that system-wide regulatory and metabolic factors may limit production outcomes, despite efficient gene upregulation.

**Combinatorial effects of CRISPRa/i multiplexing on IB/3M1B biosynthesis**

To further validate our earlier findings on the multiplexing capability of our CRISPRa system, we extended this approach in the context of metabolic engineering. Specifically, we targeted single genes with multiple gRNAs and simultaneously activated multiple genes for a subset of our candidate genes described in the previous section. Additionally, we aimed to downregulate two genes, namely _acnSP_ and _slr6040_ (Supplementary Fig. 8), by targeting their CDSs with our novel CRISPRa system. The _acnSP_ gene[40] encodes a 44-amino acid small protein previously shown to regulate the activity of the aconitase AcnB, which catalyses the conversion of citrate to isocitrate in the TCA cycle. Deletion of _acnSP_ has been reported to cause significant metabolic perturbations, notably increasing levels of pyruvate-derived metabolites, including the branched-chain amino acids L-valine and L-leucine. _slr6040_[41,42] encodes a putative two-component response regulator, potentially involved in osmotic response. Shi et al.[42] demonstrated that a Δ_slr6040_ mutant exhibited pronounced metabolic shifts, particularly in the TCA cycle, and an ~5-fold increase in pyruvate levels, particularly relevant for IB/3M1B biosynthesis. In addition to downregulating _acnSP_ and _slr6040_ individually, we also explored combining this downregulation with the upregulation of selected genes, further demonstrating the versatility of our CRISPRa system for metabolic mapping and production optimisation.

Based on our previous findings that the strongest impact on product formation was detected at day 2, we measured IB and 3M1B biosynthesis on day 2 for a combination of candidate genes (Fig. 8), all involved in pyruvate metabolism. In contrast to the increased fluorescence observed with dual targeting during tool characterisation, targeting either _pyk1_ or _pyk2_ with two gRNAs did not further enhance product formation compared to single gRNAs (Fig. 8a). However, simultaneous activation of two genes showed combinatorial effects (Fig. 8b), achieving up to a 6-fold improvement in 3M1B production compared to single targets (Fig. 8d). Furthermore, single downregulation of _slr6040_ and _acnSP_ did not significantly improve product formation under the tested conditions (Fig. 8c). Interestingly, despite these marginal improvements for single downregulation, a notable combinatorial effect was observed when combining _acnSP_ downregulation with _me_ activation (Fig. 8c, d), reaching three-fold increase in IB and 3M1B production compared to single targeting. However, multiplexing _slr6040_ repression with _me_ activation did not lead to combinatorial effects. Surprisingly, co-targeting _slr6040_ and _me_ negatively impacted product formation, particularly IB biosynthesis, suggesting an unknown metabolic or regulatory interaction in this strain.

Although synergetic effects were not observed in all multiplexed strains, the CRISPRa system functioned as anticipated, achieving robust gene repression and activation across most targets (Supplementary Fig. 9). Notably, no dilution effect was detected when simultaneously targeting multiple loci, demonstrating the system's capacity for effective multiplexed gene regulation. Unexpectedly, _me_ transcript levels were reduced when co-targeting with _acnSP_ repression, even though the identical _me_-specific gRNAs were used across all _me_-targeted strains. Additionally, in contrast to our previous findings, targeting _pyk1_ or _pyk2_ with dual gRNAs did not further enhance target activation, aligning with the product formation levels detected.

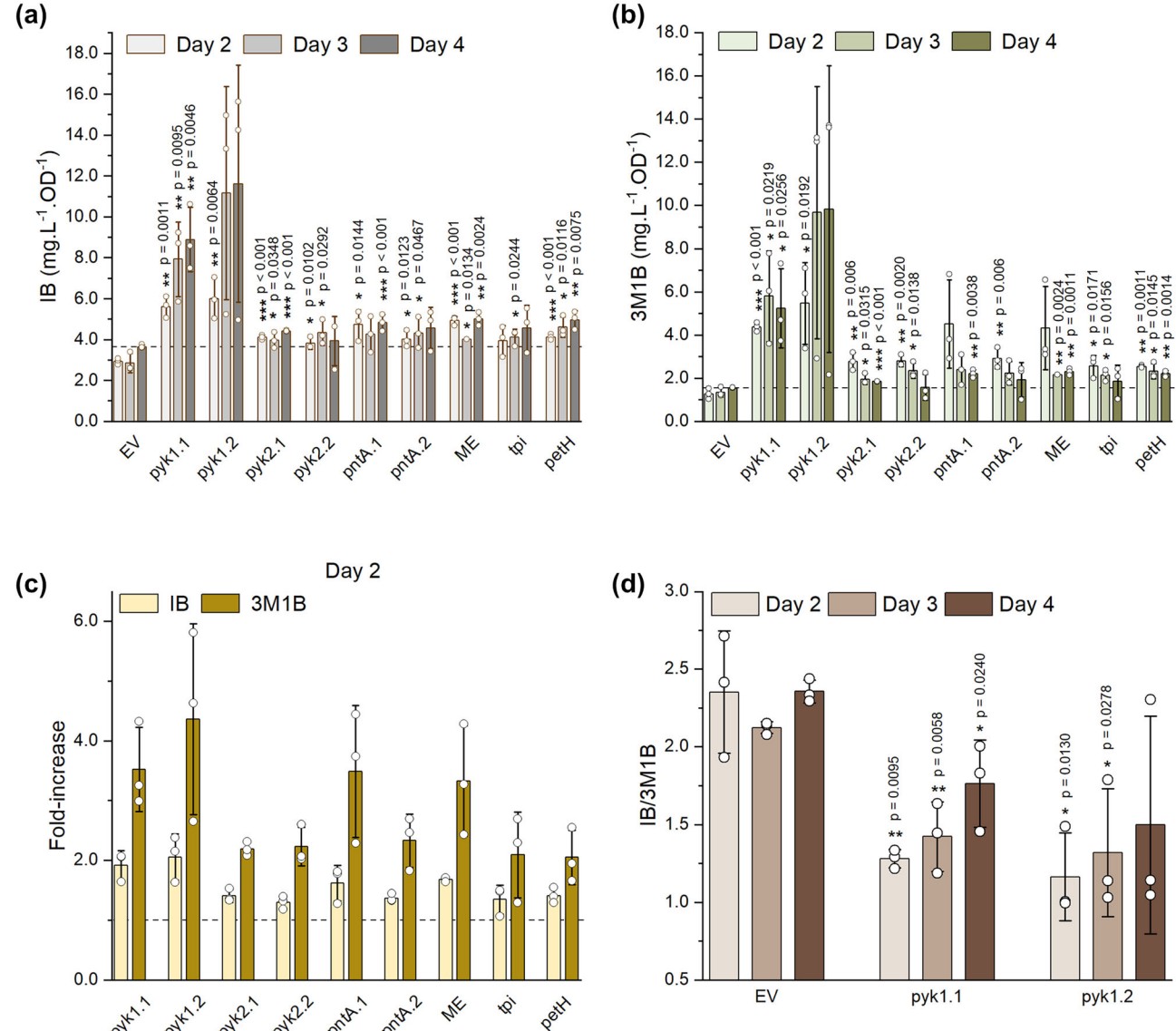

**Fig. 7 | Isobutanol and 3-methyl-1-butanol production in response to CRISPRa targeting of select genes involved in pyruvate formation and NADPH regeneration. a** Isobutanol titres on days 2, 3, and 4. **b** 3-methyl-1-butanol titres on days 2, 3, and 4. **c** Fold-increase of isobutanol and 3-methyl-1-butanol relative to the negative control (EV) on day 2. **d** IB/3M1B ratio for *pyk1* activation. gene.1 and gene.2 refers to gRNAs #1 and #2, respectively, when applicable. EV negative control—HX11 expressing plasmid pBB_CA, IB isobutanol, 3M1B 3-methyl-1-butanol. Error bars indicate standard deviation ($n = 3$). $p$ value representation: * < 0.05; ** < 0.01; *** < 0.001.

## Discussion

In this study, we developed a dCas12a-mediated CRISPRa system for targeted gene upregulation in *Synechocystis*, significantly expanding the toolkit available for cyanobacteria. Our initial characterisation experiments revealed that the designed tool exhibits a flexible editing window, with minimal differences between gRNAs targeting an optimal region approximately -100 to -200bp upstream of the TSS. We hypothesise that this window is determined by the spatial constraints required for direct interactions between the dCas12a-tethered SoxS and RNA polymerase, which are required for successful activation[11]. As a previous study[19] showed that SoxS-dependent gene expression was highly sensitive to the distance between the TSS and the SoxS binding site in its native context, the positioning of the dCas12a binding site relative to the TSS, dictating the positioning of SoxS, likely determines activation efficiency. However, in contrast to previously reported Cas9-based CRISPRa systems in other prokaryotes, which demonstrate significant changes in activation levels at a single-nucleotide resolution[20,24], our system offers greater flexibility, particularly advantageous for targeting endogenous promoters with limited PAM sequences. However, despite this increased targeting flexibility, efficient activation of certain endogenous genes may still be constrained by the absence of suitable PAM sequences within the optimal distance from the TSS—an inherent limitation of CRISPR-based approaches[43]. Furthermore, we observed increased activation at two separate loci when targeting the non-template strand opposite the promoter direction, providing an additional strategy to fine-tune activation levels for future applications. Consistent with previous studies in other prokaryotes[20,24] despite *Synechocystis* unique regulatory mechanisms, activation levels were found to be inversely correlated with the baseline expression levels of the target promoter, with weaker promoters showing a higher fold-increase than strong promoters. While this characterisation provides important insights into the tool's behaviour, other factors, such as the DNA sequence between the gRNA target site and the promoter[20,24], have been shown to influence activation levels but were not explored here. Future studies could thus investigate these parameters to refine CRISPRa efficiency in *Synechocystis* and reach maximal

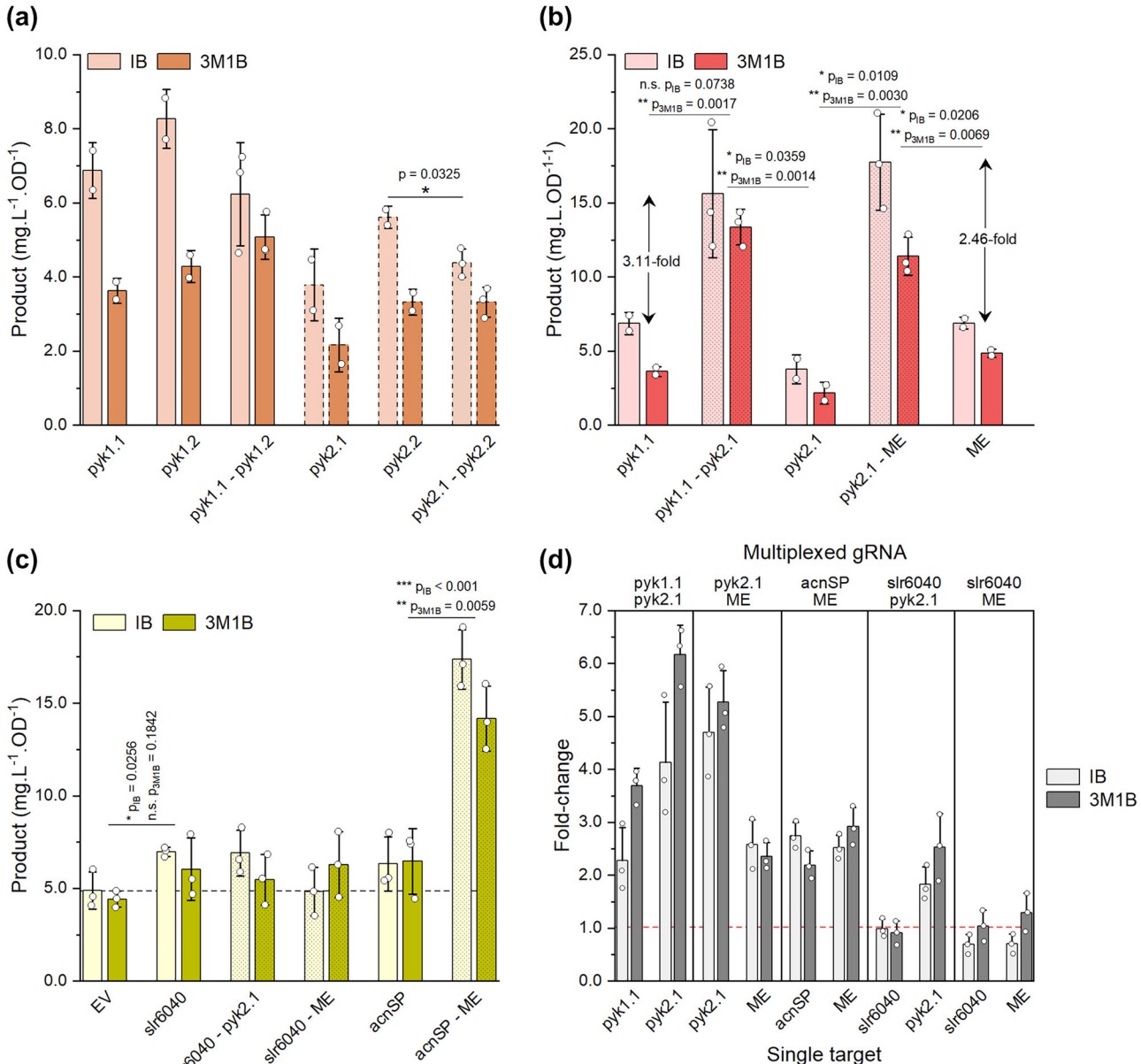

**Fig. 8 | IB/3M1B biosynthesis in response to CRISPRa multiplexing for simultaneous down- and upregulation of several target genes. a** Targeting of *pyk1* or *pyk2* with two gRNAs. pyk2 strains are represented with bars with a dashed outline. **b** Simultaneous activation of *pyk1* and *pyk2* or *pyk2* and *me*. Fold-improvement are shown as the mean fold-change in IB and 3M1B production. **c** Downregulation of *slr6040* and *acnSP* using single or multiplexed gRNAs. **d** Fold-change of IB and 3M1B biosynthesis in multiplexed strains compared to single gRNA-targeted strains. EV negative control—HX11 expressing plasmid pBB_CA, IB isobutanol, 3M1B 3-methyl-1-butanol. Error bars represent standard deviation (single activation: $n = 2$; single repression and multiplexed targeting: $n = 3$). $p$ value representation: n.s. > 0.05; * < 0.05; ** < 0.01; *** < 0.001.

activation for synthetic genetic constructs. Notably, despite characterisation efforts, the fold-change in gene expression achieved in *Synechocystis* was significantly lower than in other organisms, where increases of ~50–100-fold[20,24] have been reported. Similar activation levels were observed in *S. elongatus* PCC 7942 using a dCas9-based system, with fold-changes ranging from 2 to 10[19], indicating that a limited dynamic range may be an inherent feature of cyanobacterial transcriptional regulation. An early transcriptomics analysis[44] under stress conditions in *Synechocystis* suggested that its native transcriptional regulatory mechanisms exhibit a narrow dynamic range, further supported by studies focusing on specific stress conditions[45,46] or mutant strains[47]. As a result, it is possible that fold-activation may not reach the levels observed in other prokaryotes, even with additional tool optimisation. Additionally, given that *Synechocystis* is highly polyploid[48], the high number of chromosomal copies may influence

activation efficiency, potentially reducing the overall expression fold-increase. Notably, chromosomal copy number is highly dynamic, varying with environmental conditions, growth stages, and even amongst individual cells within the same population[49], suggesting that activation efficiency may fluctuate over time and between cells. Since our study assessed tool efficiency at the population level, we may overlook potential heterogeneity in gene activation. Single-cell approaches could provide deeper insights into these variations. Nevertheless, despite lower fold-activation, CRISPRa remains a powerful and versatile tool in *Synechocystis*, as demonstrated by its applicability for metabolic engineering in this study.

We further investigated the potential of our CRISPRa system for metabolic engineering by activating the *kivD*[S286T] gene[29] at the $P_{trc}$ promoter to enhance IB and 3M1B production. Using both single and multiplexed gRNAs, we upregulated 1-to-3 copies of *kivD*[S286T] in three separate

background strains. While transcript levels of $kivD^{S286T}$ were successfully increased in most strains, this strategy only resulted in marginal improvements in IB and 3M1B biosynthesis. As we did not investigate translational regulation in our experiments, potential regulatory mechanisms may have limited protein accumulation, although the engineered strains contained a bicistronic design commonly used for strong translation of heterologous proteins. While KivD has previously been identified as a critical bottleneck in this biosynthetic pathway[29,30], additional regulatory mechanisms, specifically those impacting intermediate availability, may be limiting IB and 3M1B formation. The native regulation of the branched-chain amino acid (BCAA) biosynthesis in *Synechocystis* remains largely unexplored but directly influences IB/3M1B formation. In other organisms, tight transcriptional and enzymatic control, with negative feedback loops and allosteric inhibition, respectively, has been shown to limit intermediate formation in this metabolic pathway[50], which may explain the modest improvements observed in our experiments. Notably, an early study reported that BCAA concentrations influence *kivD* expression and potentially its enzymatic activity in *Lactococcus lactis*[51], further highlighting the intricate relationship between BCAA biosynthesis and IB/3M1B formation. We, thus, suggest that future metabolic engineering efforts should focus on addressing these potential regulatory constraints to substantially enhance cyanobacterial IB/3M1B biosynthesis. Furthermore, our findings suggest that $kivD^{S286T}$ upregulation was primarily beneficial for IB/3M1B production at early timepoints, while its impact diminished at later growth stages. This pattern indicates the presence of unknown regulatory and metabolic constraints that limit IB/3M1B biosynthesis. Future research should incorporate multi-omics analyses across different timepoints to identify such underlying mechanisms. KivD activity has been shown to depend on pH and ion concentrations[52], raising the possibility that intracellular conditions become less favourable for IB/3M1B formation as the culture progresses. Notably, *Synechocystis* cultures have been observed to rapidly alkalinise[53], while KivD activity significantly declines at pH levels above 7.5. Moreover, drastic transcriptome-wide changes occur across different growth phases in *Synechocystis*[54], which may reduce the effectiveness of our CRISPRa system at later stages. Controlling this transcriptional rewiring to maintain strong activation may be challenging, as it likely involves complex regulatory networks that dynamically adapt to cellular and environmental conditions. Unexpectedly, we observed differing behaviour of identical gRNAs in separate background strains, as exemplified by gRNAs NS1 and NS1-ddh in HX11 and HX51-derived strains. While the precise cause of this variation remains unclear and requires additional work to elucidate, we hypothesise that this behaviour is not directly related to the CRISPRa system itself. Instead, this may be due to mutation or recombination events occurring in different background strains. Genetic instability has previously been recognised as a key challenge in cyanobacterial producer strains[55,56], although recent work in *S. elongatus* PCC 7942 demonstrated that targeted genetic engineering was required to increase mutation rates[57]. In addition to mutational events in genes involved in product formation, uncontrolled phenotypic instability[58], may also contribute to the differential behaviour observed. This phenomenon remains poorly understood, highlighting the need for a more comprehensive system-wide understanding of producer strain dynamics. Such genetic or phenotypic instability may contribute to the slight reduction of IB/3M1B formation observed in some strains, particularly prevalent in the HX51 background. Given that HX51 strains harbour three copies of an identical $kivD^{S286T}$ construct, intra- and interchromosomal recombination—an underexplored phenomenon in cyanobacteria[59]—could potentially rearrange heterologous genetic elements, affecting CRISPRa-mediated gene upregulation. Additionally, metabolic burden from multiple antibiotic resistance, increased carbon flux towards IB/3M1B formation, and high proteome allocation to KivD may further exacerbate strain instability, contributing to the observed production decline. Further investigation is required to elucidate the mechanisms underlying the reduced production in some HX51-activated strains, particularly the potential interplay between genetic instability and metabolic constraints.

Furthermore, we targeted six native genes associated with pyruvate formation and NADPH regeneration with our CRISPRa system, facilitating rapid target mapping to identify metabolic processes relevant for IB and 3M1B biosynthesis. With the exception of gRNA pyk2.1, which did not result in successful activation (Supplementary Fig. 7), all candidate genes were significantly upregulated, validating the CRISPRa system's applicability for precise upregulation of endogenous genes. However, as observed in other organisms[19], we noted variability in activation levels across different targets. This variability may be attributed to target-specific factors, such as basal expression levels, constraints imposed by native regulatory mechanisms, and differences in gRNA binding affinity and efficiency. Since the primary objective of this experiment was to assess the applicability of our CRISPRa system for target mapping, we did not explore strategies to maximise activation levels for each selected target. However, we anticipate that activation levels could be further improved by, for instance, screening multiple gRNAs if maximal activation is required for future applications. Additionally, activation levels did not directly correlate with fold-changes in product formation, indicating that a complex and dynamic regulation on a system-wide level influences compound production. While most targets led to increased product formation, upregulation of *pyk1* had the most pronounced effect on IB/3M1B levels, resulting in up to a four-fold increase on day 2. Notably, gene activation of some targets, such as *tpi* or *pntA*, only showed transient effects on compound bioproduction. We hypothesise that this phenomenon is due to native regulatory mechanisms governing these reactions, particularly important during early growth phases. These findings not only demonstrate the potential of the CRISPRa system for metabolic mapping but also provide insights into the metabolic processes influencing IB/3M1B biosynthesis in *Synechocystis*, which may extend to other compounds, such as isobutene[7], originating from 2-ketoacid intermediates. Differential improvements observed between *pyk1* and *pyk2* activation suggest distinct regulatory mechanisms controlling these two targets, consistent with previous studies[31,60]. While transcriptional regulation of both *pyk1* and *pyk2* is responsive to glucose concentrations[61], Pyk1 is inhibited by high ATP levels[60], suggesting that it is inactive during photoautotrophic growth in response to high ATP levels generated during photosynthesis[32]. Conversely, Pyk2 regulation is linked to the accumulation of sugar phosphates[31], hypothesised to be essential for sugar catabolism under hetero- or mixotrophic conditions. Despite these reported regulatory mechanisms, the high activation levels achieved for both genes suggest that regulation governing these enzymes was less stringent in our experiments, allowing significant upregulation through CRISPRa targeting. Additionally, we observed enhanced IB/3M1B biosynthesis when upregulating *me*. Malate oxidation by ME has previously been reported as a key route for pyruvate formation[62], potentially bypassing pyruvate kinase, which potentially contributed to elevated pyruvate availability in our experiments, improving IB/3M1B biosynthesis. While a previous study showed that ME overexpression did not enhance ethanol production under photoautotrophic conditions[63], it did increase D-lactate biosynthesis under dark anoxic conditions[33]. Although $^{13}C$ flux analysis has previously reported that ME is active under light conditions[62], the role of light regulation in ME function remains unclear. Therefore, the improved IB/3M1B production observed in response to *me* upregulation could be attributed to both increased pyruvate formation and additional metabolic effects. As ME-dependent oxidation of malate derived from the TCA cycle has been shown to generate NADPH[32], we cannot exclude the possibility that *me* activation contributed to NADPH regeneration, important for IB/3M1B biosynthesis. *tpi* upregulation also led to modest improvements in IB/3M1B production, in accordance with previous results[5]. Collectively, our results further support that pyruvate availability represents a key bottleneck in IB/3M1B biosynthesis and requires additional engineering efforts for sustained improvement of IB/3M1B bioproduction in *Synechocystis*. Interestingly, *pntA* activation led to modest improvements, despite a previous report suggesting its importance primarily under mixotrophic conditions[37]. Similarly, targeting *petH* moderately enhanced compound bioproduction, with sustained effects throughout the experiment. These findings report a

strategy for NADPH regeneration in the context of producing pyruvate-derived compounds in *Synechocystis*, highlighting that improving NADPH regeneration could be a promising avenue for increasing IB/3M1B production and warrants further investigation. Collectively, our approach successfully identified key metabolic reactions that constrain product formation. Further target exploration and validation, as well as engineering efforts, such as permanent genome editing via strategies like promoter swapping, could significantly advance IB/3M1B bioproduction in *Synechocystis*. Notably, while our CRISPRa system offers a powerful tool for rapid target testing, the compact genome of *Synechocystis* poses challenges in achieving highly specific activation in some cases, as targeting bidirectional promoter regions or operons cannot always be avoided, a challenge previously reported during the generation of a CRISPRi library[16]. For example, the *pyk2* promoter region, targeted with the designed gRNAs, overlaps with the promoter region of *slr1362*, a gene of unknown function. Consequently, we cannot entirely exclude the possibility that the observed phenotypic changes are solely attributed to *pyk2* upregulation and may instead be influenced by the unintended activation of adjacent genes. Similarly, given the inherent potential for off-target effects in CRISPR-based systems, a comprehensive off-target analysis would be valuable to determine whether activation of additional genes contributed to the observed phenotypes.

We further investigated multiplexing our CRISPRa system for simultaneous targeted gene activation and repression. Our findings confirmed that strong repression could be achieved by targeting within the CDS, effectively blocking transcript elongation. Additionally, no dilution effects were observed during multiplexing, enabling strong activation of multiple targets. However, contrary to our GFP activation results, dual gRNAs did not further enhance activation levels of *pyk1* or *pyk2* or improve compound production. This may partly result from reaching an expression threshold for these genes, although additional factors, such as target site accessibility, may also play a role. To further assess whether dual targeting is beneficial, activation of additional endogenous genes with 2 gRNAs should be explored. Interestingly, significant increases in product formation, correlated with elevated transcript level, were achieved when co-targeting *pyk1* and *pyk2* or *pyk2* and *me*, demonstrating a synergetic effect in these strains. While both *acnSP* and *slr6040* repression was strong, only marginal improvements in product formation were observed. A previous study found that a Δ*acnSP* mutant[40] yielded an approximately two-fold increase in valine and leucine biosynthesis under higher light conditions (100 μmol photons m$^{-2}$ s$^{-1}$), while our experiments were performed at 50 μmol photons m$^{-2}$ s$^{-1}$ for consistency with earlier IB/3M1B studies[4,18,30]. This light intensity difference may account for the limited product increase observed in our experiments. Although single *acnSP* targeting had minimal impact, co-targeting *acnSP* with *me* substantially improved IB/3M1B formation. Surprisingly, *me* transcript levels were reduced in the co-targeted strain. Interestingly, a 1.5-fold increase in intracellular malate was observed in the Δ*acnSP* mutant[40], which would increase substrate availability for ME. This unexpected behaviour warrants further investigation as unidentified regulatory and metabolic interactions involving AcnSP may influence ME expression or activity. Similarly, strong repression of *slr6040* had limited impact on product formation, despite a previous study reporting a 4.83-fold increase in pyruvate levels in a Δ*slr6040* mutant[42]. Since the exact signal(s) and output of this two-component response regulator are unknown, its lack of effect on product formation remains speculative. Interestingly, co-targeting *slr6040* with activated genes negated the positive impact of gene activation in both multiplexed strains, suggesting that *slr6040* repression leads to system-wide metabolic disturbances. Expanding this multiplexing approach to include targets beyond those directly linked to pyruvate formation would provide valuable insights into regulatory and metabolic processes influencing production. Notably, the results from multiplexed and single-target experiments highlight the need for an improved understanding of system-wide regulatory mechanisms and their impacts in the context of metabolic engineering for compound bioproduction. For instance, shifts in IB/3M1B ratio were observed across several strains when targeting genes related to

pyruvate formation, in accordance with a previously reported phenotype[18]. This warrants extensive omics analysis to uncover associated regulatory mechanisms.

Our CRISPRa system significantly expands the genetic toolbox available for *Synechocystis*, enabling targeted and efficient gene activation for both heterologous and endogenous genes. Our results on IB/3M1B production further highlight its value for rapid target mapping and multiplexed activation for metabolic engineering strategies. While direct comparisons of production titres across studies remain challenging due to variations in experimental conditions[64], such as light intensity and culture setup, our approach achieved up to a four-fold increase in product formation with single-gene activation and up to a six-fold increase with multiplexed targeting. These results highlight key metabolic pathways limiting IB/3M1B biosynthesis. Beyond identifying metabolic reactions influencing compound production, this tool can be employed for promoter optimisation within biosynthetic pathways using randomised gRNAs, a strategy previously demonstrated for violacein biosynthesis[22]. Additionally, CRISPRa-based regulation can be integrated with other regulatory elements to build complex genetic circuits[65,66], allowing cellular signal computing. The potential of this CRISPRa system extends beyond metabolic engineering, offering an interesting opportunity for upregulation of large operons or silent genes[66,67] relevant for fundamental studies. Similarly, genome-wide CRISPRa studies could also be particularly insightful for both applied and fundamental research, enabling the identification of target genes for improved production and characterise gain-of-function phenotypes under various environmental conditions[68], although CRISPRa libraries are still limited in prokaryotes. However, gRNA design remains a critical factor for precise and efficient gene activation. Recent algorithms have refined gRNA prediction[69], although it remains difficult to prevent off-target effects. This challenge will be particularly relevant for CRISPRa libraries in *Synechocystis* as its compact genome contains numerous overlapping promoters, as noted in our study. However, as the CRISPR field advances, new Cas enzymes and engineered variants[27,70] expand targeting capabilities, increasing genome coverage and further broadening applications of CRISPRa systems. Integrating these approaches with our CRISPRa system promises valuable advancements for next-generation metabolic engineering in *Synechocystis*.

## Methods
### Strains and growth conditions
*Escherichia coli* DH5α Z1 (Invitrogen) or T7 Express (NEB) were cultivated at 37 °C in LB medium supplemented with 50 μg mL$^{-1}$ kanamycin or 50 μg mL$^{-1}$ spectinomycin, when applicable. Background strains of *Synechocystis* sp. PCC 6803 used in this study are listed in Supplementary Table 1. *Synechocystis* strains were routinely grown in BG11 medium[71], supplemented with the appropriate antibiotics (kanamycin: 50 μg mL$^{-1}$ (single antibiotic selection) or 25 μg mL$^{-1}$ (multiple antibiotic selection); spectinomycin: 50 μg mL$^{-1}$ (single antibiotic selection) or 25 μg mL$^{-1}$ (multiple antibiotic selection); chloramphenicol: 25 μg mL$^{-1}$ (single antibiotic selection) or 10 μg mL$^{-1}$ (multiple antibiotic selection); erythromycin: 25 μg mL$^{-1}$). Cultures were incubated at 30 °C, 120 rpm, and 50 μmol photons m$^{-2}$ s$^{-1}$. Seed cultures from cryostocks were inoculated in six-well plates (Sarstedt) and grown under 30 μmol photons m$^{-2}$ s$^{-1}$ illumination. For fluorescence-based tool characterisation, biological replicates were cultivated in 24-well plates (Sarstedt), with 3 glass beads per well to maximise agitation. Tool stability experiments (Supplementary Fig. 2) were conducted in biological triplicate in 100-mL Erlenmeyer flasks containing 25 mL of BG11. For IB and 3M1B experiments, cultivation conditions are described in the section "IB and 3M1B quantification". CRISPRa induction was initiated by the addition of 3 mM of L-rhamnose (Sigma-Aldrich) during culture inoculation, unless stated otherwise. Growth profiles were monitored by measuring the optical density at 750 nm using a Hidex Plate Chameleon plate reader. All experimental cultures were inoculated at a starting OD$_{750}$ of 0.1.

In all experiments, the negative control, referred to as EV, corresponds to the respective background strain carrying plasmid pBB_CA, which

encodes the dCas12a-SoxS fusion and a CRISPR array without a targeting gRNA, both regulated by the rhamnose-inducible system. Unless otherwise stated, EV controls were subjected to the same treatment as the CRISPRa-targeted strains.

## Plasmid construction

All plasmids used in this study are listed in Supplementary Table 2. Oligonucleotides were purchased from Integrated DNA Technologies (IDT), with sequences provided in Supplementary Table 3. To construct the initial CRISPRa vector, pBB_CA (Supplementary Fig. 1), a codon-optimised version of *soxS*[R93A] for *Synechocystis* (Supplementary Table 4) was synthesised by IDT and inserted into the pBB_dCas12a vector[18] using NEBuilder® HiFi DNA assembly (NEB). For gRNA cloning, complementary 5'-phosphorylated oligonucleotides were purchased (IDT) with the following extensions: AGAT (forward primer) and AGAC (reverse primer), as previously described[72]. Primer pairs were annealed in nuclease-free Duplex buffer (IDT) by combining 2 μL of each primer (100 μM) with 46 μL Duplex buffer and incubating the reaction for 2 min at 94 °C followed by gradual cooling to room temperature. The resulting annealed gRNAs were cloned into AarI-digested pBB_CA. pBB_CA and its gRNA-containing derivatives featured the broad-host-range RSF1010 replicon and mobilisation region[73]. For genomic integration of GFP into the NS1 (*slr0168*) site[74], the GFP CDS was amplified from pEB1-wtGFP[75] (purchased from Addgene; accession number #103978) using Phusion™ Hot Start II DNA Polymerase (ThermoFisher) and subsequently fused to the respective promoters via overlapping PCR. The resulting constructs were cloned into the P3 vector[4] using EcoRI and NotI restriction enzymes (ThermoFisher Scientific). All plasmids were confirmed by Sanger sequencing (Eurofins).

## *Synechocystis* natural transformation and conjugation

Integrative plasmids were transformed into *Synechocystis* via natural transformation as previously described[30]. Briefly, cultures were grown to an $OD_{750}$ of ~1.0, centrifuged at $6000 \times g$ for 10 min, washed twice in BG11 medium, and resuspended at a final cell density of $1 \times 10^9$ cells mL$^{-1}$. 400 μL of cell suspension was incubated with 4 μg of plasmid DNA for 4–5 hours at 30 °C, 120 rpm and 50 μmol photons m$^{-2}$ s$^{-1}$ and plated on nitrocellulose membranes placed on non-selective BG11 agar plates. After 20–24 hours, the membranes were transferred to selective plates with the appropriate antibiotic(s) and incubated at 30 °C and 50 μmol photons m$^{-2}$ s$^{-1}$ until colonies appeared (7–10 days). Colonies were restreaked onto selective BG11 agar plates and screened for positive genomic integration by PCR with DreamTaq DNA polymerase (ThermoFisher). Selected colonies were inoculated in selective BG11 medium in six-well plates and propagated until fully segregated. Strains ddh_kivD, HX11 (two copies of *kivD*[S286T]), and HX51 (three copies of *kivD*[S286T] inserted at the *ddh*, NS1, and *sll1564*[4] sites) were constructed via homologous recombination for other studies as described elsewhere[18,30].

For self-replicating plasmids, a triparental conjugation method was used, as previously described[30]. Briefly, the cargo *E. coli* strain carrying the target plasmid and the helper strain[76] HB101 containing the pRL443-Amp$^R$ plasmid were grown overnight at 37 °C in LB, supplemented with 50 μg mL$^{-1}$ kanamycin and 100 μg mL$^{-1}$ ampicillin, respectively. After overnight growth, 1 mL of each *E. coli* strain and the recipient *Synechocystis* strain (at $OD_{750}$ = ~ 1.0) was centrifuged at $3000 \times g$ for 5 min and resuspended in fresh LB or BG11. 500 μL of cell suspension of cargo strain and helper strain were combined and washed twice in LB. Concurrently, *Synechocystis* cells were washed twice in BG11. 50 μL of *Synechocystis* cell suspension was added to the *E. coli* mixture, incubated for 1.5–2 hours at 30 °C, 120 rpm and 50 μmol photons m$^{-2}$ s$^{-1}$ and plated on nitrocellulose membranes on non-selective BG11 plates. After 20–24 hours, the membranes were transferred to selective BG11 plates and incubated at 30 °C and 50 μmol photons m$^{-2}$ s$^{-1}$ until colonies formed (5–7 days). Single colonies were restreaked onto selective BG11 plates and analysed by colony PCR. Positive transformants were inoculated in six-well plates in selective BG11 and used for subsequent experiments.

## Fluorescence measurements

GFP fluorescence was quantified with a Hidex Plate Chameleon plate reader with an excitation at 390 nm and emission at 500 nm. Optical density was simultaneously measured at 750 nm. Fluorescence measurements were normalised to $OD_{750}$.

## RNA extraction and RT-qPCR

For RNA extraction, cells were harvested at $3000 \times g$ for 10 min and washed twice with sterile $H_2O$. Cells pellets were stored at $-80$ °C until further processing. Pellets were thawed and resuspended in 500 μL Trizol (Sigma-Aldrich). 0.2 g of acid-washed glass beads (Sigma-Aldrich) were added to cell suspensions to facilitate cell disruption. Cell lysis was carried out using a Precellys-24 bead beater (Bertin Instruments) with 3 consecutive cycles of 30 sec at $3000 \times g$ followed by 2 min on ice between cycles. To remove cell debris and glass beads after lysis, samples were centrifuged at $12000 \times g$ at 4 °C for 10 min, and supernatants were collected. RNA was isolated using the Direct-zol™ RNA Microprep kit (Zymo Research) following the manufacturer's protocol, excluding the DNase treatment step. RNA concentrations were determined using a Nanodrop™ 2000 spectrophotometer (Thermo Scientific), and 1 μg of RNA was treated with DNase I (ThermoFisher) in two consecutive reactions to eliminate potential genomic DNA. The resulting samples were then purified a second time with the Direct-zol™ RNA Microprep kit to increase sample purity. For cDNA synthesis, 150 ng of RNA was used as template for the iScript™ cDNA synthesis kit (BIO-RAD) according to the manufacturer's recommendations. Real-time quantitative PCR (RT-qPCR) was performed on 5 ng of cDNA with iTaq Universal SYBR Green Supermix (BIO-RAD) in a CFX Connect Real-Time PCR system (BIO-RAD). The *rnpB* gene, encoding the RNA subunit of ribonuclease P, served as the reference gene, with primers previously described[77]. Other primer sequences are provided in Supplementary Table 3. Each sample was analysed in three technical replicates, and relative gene expression levels were calculated using the $2^{-\Delta\Delta CT}$ method[78].

## gRNA design for target mapping

To effectively target candidate genes, gRNAs were designed based on insights gained from our characterisation experiments on GFP activation. The gRNAs were specifically directed to the non-template strand within the optimal targeting window positioned between $-100$ and $-150$ bp upstream of the TSS. TSS were identified using data from a previously published transcriptomics analysis[44].

## IB and 3M1B quantification

For production experiments, seed cultures were inoculated in 6-well plates in selective BG11 supplemented with appropriate antibiotics and incubated at 30 °C and 30 μmol photons m$^{-2}$ s$^{-1}$. 25 mL of BG11 supplemented with 50 mM NaHCO$_3$ (Sigma-Aldrich), and 3 mM L-rhamnose were inoculated at $OD_{750}$ = 0.1 from seed cultures in BioLite 25 cm$^2$ plug-sealed tissue culture flasks (ThermoFisher Scientific), as previously described[30]. The cultures were horizontally shaken at 120 rpm under 50 μmol photons m$^{-2}$ s$^{-1}$ illumination at 30 °C. Every second day, 2 mL of cultures were replaced by 2 mL of BG11 supplemented with 500 mM NaHCO$_3$, 3 mM L-rhamnose, and appropriate antibiotics. $OD_{750}$ was monitored daily through measurement with a Hidex Plate Chameleon plate reader, and product formation was quantified on days 3, 4, 6, 8, and 10 for experiments with multiple *kivD*[S286T] copies or on days 2, 3, and 4 for target mapping. Cultures were performed in biological triplicates.

IB and 3M1B were extracted from culture samples using a dichloromethane extraction method previously described[30]. Both products were quantified by gas chromatography using a PerkinElmer GC 580 system equipped with a flame ionisation detector) and an Elite-WAX Polyethylene Glycol Series Capillary column (30 m × 0.25 mm × 0.25 μm, PerkinElmer) with nitrogen as the carrier gas at a rate of 10 mL min$^{-1}$. The injector and detector temperatures were set at 220 °C and 240 °C, respectively. The GC results obtained were analysed using TotalChrom Navigator version 6.3.2.

**Article**

## Statistics and reproducibility

All data are presented as the mean value ± standard deviation (SD). Experiments for fluorescence-based tool characterisation had a sampling size of $n \geq 5$. All IB/3M1B experiments were performed in biological triplicates ($n = 3$). Statistical significance was determined using an unpaired two-tailed $t$ test. Relevant $p$ values are reported on the respective figures (ns (not significant) > 0.05; * < 0.05; ** < 0.01; *** < 0.001). All data supporting Figs. 2–8, as well as Supplementary Figs. 1–6 and 8, are available in Supplementary Data 1.

## Reporting summary

Further information on research design is available in the Nature Portfolio Reporting Summary linked to this article.

## Data availability

The main data supporting the findings of this study are available in the article, its Supplementary Information and Supplementary Data 1.

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

## Acknowledgements
This research was financially supported by Formas—A Swedish Research Council for Sustainable Development (project no. 2021-01669 to K.S.).

## Author contributions
B.B.: conceptualisation; data curation; formal analysis; investigation; methodology; validation; visualisation; writing–original draft; writing–review & editing. H.X.: development of background strains ddh_kivD, HX11, and HX51; technical support for IB/3M1B quantification; writing–review & editing. P.L.: writing–review & editing. K.S.: funding acquisition; project administration; resources; supervision; writing–review & editing. All authors read and approved the final version of the manuscript.

## Funding

## Competing interests
The authors declare no competing interests.
