## [Transparent Peer Review file · Communications Biology]

Development of a CRISPR activation system for targeted gene upregulation in *Synechocystis* sp. PCC 6803

Corresponding Author: Professor Karin Stensjö

Version 0:

Reviewer comments:

Reviewer #1

(Remarks to the Author)

The manuscript presents the development and application of a CRISPR activation (CRISPRa) system for the cyanobacterium *Synechocystis* sp. PCC 6803. This system enables robust, multiplexed gene activation of both endogenous and heterologous targets. The authors employed this approach to identify and upregulate genes influencing isobutanol (IB) and 3-methyl-1-butanol (3M1B) production. Their results demonstrate up to a fourfold increase in compound production through individual and multiplexed gene activation, shedding light on complex regulatory interactions governing biosynthesis. The study provides a valuable tool for high-throughput screening, metabolic pathway optimization, and functional genomics, representing a notable advancement in genetic engineering for *Synechocystis*.

Since CRISPRa has been successfully applied in various organisms, it would strengthen the study to discuss both the common challenges encountered when implementing CRISPRa across different systems and the specific challenges unique to *Synechocystis* or cyanobacteria. Addressing these aspects would help clarify whether adaptations were necessary for the system to function effectively in *Synechocystis* and how these insights might inform future applications in related organisms. Additionally, given that cyanobacteria have been used previously to produce IB and 3M1B from CO₂, comparing the productivities achieved in this study with those reported in prior work without CRISPRa would help contextualize its impact on production efficiency.

Reviewer #2

(Remarks to the Author)

1. Brief summary of the manuscript

This manuscript presents a CRISPR activation (CRISPRa) system for *Synechocystis* sp. PCC 6803, enabling targeted gene upregulation for metabolic engineering. The authors demonstrate its effectiveness by enhancing biofuel production, showing that activating genes like *pyk1* increases isobutanol and 3-methyl-1-butanol yields. Interestingly, gene activation did not always correlate with metabolite production, suggesting complex regulatory effects. The study provides a useful tool for strain optimization.

2. Overall impression of the work

This study represents a valuable contribution to the genetic toolkit available for cyanobacterial engineering, offering a CRISPRa-based approach for targeted gene activation. The system has potential applications not only in metabolic engineering, such as high-throughput screening and pathway optimization, but also in fundamental studies aimed at understanding the regulation of metabolic pathways in *Synechocystis*. The authors provide a thorough characterization of the CRISPRa system developed, demonstrating the advantages of using dCas12a for precise gene upregulation. The work is scientifically sound, with well-designed experiments and clear methodology. While further validation under diverse conditions and comparisons with existing approaches would strengthen its impact, the study presents a promising and useful tool for the field.

Overall, the manuscript is of high quality and provides meaningful insights. I support its publication, pending minor revisions to improve clarity and address the points raised.

3. Specific comments, with recommendations for addressing each comment

Lines 46–48: The statement regarding the limited availability of genetic tools in *Synechocystis* should be nuanced. While

improvements are still needed, *Synechocystis* is one of the most extensively studied cyanobacteria, with a well-established genetic toolkit. Consider rephrasing to reflect this more accurately.

Lines 113–116: The authors report that gRNAs positioned between -97 and -156 yielded the highest fluorescence levels, suggesting this region as optimal for gene activation via dCas12a-SoxS in *Synechocystis*. Could the authors propose a mechanistic explanation for why this region is particularly effective? For instance, could it be related to RNA polymerase accessibility or the presence of specific regulatory elements?

Figure 2: The description of the negative control should be expanded. Please clarify its exact role in the experiment and how it validates the observed effects.

Lines 177–178: "At later time points, IB and 3M1B levels became comparable between CRISPRa-targeted and non-targeted strains." Could the authors provide a hypothesis for this convergence in production levels? Potential explanations regarding metabolic feedback, resource allocation, or regulatory mechanisms would be valuable.

Figure 4: The figure legend should be revised to provide a more complete and clearer explanation of the data presented. Currently, it lacks sufficient detail to fully interpret the results. Specifically, the labels for the different experimental conditions need to be clarified, and the relationship between the data in the figure and the corresponding experimental setup should be explicitly defined. This will improve the readability and ensure that the figure is easily understood by readers.

Lines 190–192: The manuscript describes a modest reduction in IB and 3M1B production in some CRISPRa-targeted strains, particularly in HX51 derivatives. Can the authors suggest a possible explanation for this decrease? Could it be linked to metabolic burden, competition for cellular resources, or unintended regulatory effects?

Lines 197–199: The lower activation levels and delayed growth observed in HX11-derived strains (Supplementary Fig. 3) are attributed to an unidentified phenotype in the HX11 background. Could the authors provide further details or specific examples of potential factors influencing this response?

Lines 187–188: The phrase "dual targeting of the NS1 and *ddh* loci resulted in opposite effects on IB and 3M1B production in HX11" is unclear. If the observed effect was a decrease in production, Figure 4B does not seem to reflect this. Please clarify the intended interpretation.

Lines 189–190: If gRNAs were designed upstream of the promoter, please specify the exact position for each gene and its respective genetic background to facilitate reproducibility.

Lines 384–386: "All candidates demonstrated robust activation, validating the CRISPRa system's applicability for precise upregulation of endogenous genes."

If referring to Supplementary Figure 6 (qPCR results), a fold-change of 1 does not indicate an actual change in gene expression. This statement should be nuanced accordingly. Additionally, it is important to differentiate between fold-change in IB and 3M1B production and direct upregulation of target genes. A more appropriate approach might be to compare expression levels with a non-activated control.

General Considerations:

Negative controls: The description of negative controls should be more detailed for each experiment to improve clarity and ensure reproducibility.

Vector design: Adding a schematic representation of the vector containing multiple gRNAs would enhance the understanding of the experimental setup.

Reviewer #3

(Remarks to the Author)

Version 1:

Reviewer comments:

Reviewer #1

(Remarks to the Author)

The authors have suitably addressed my concerns.

Reviewer #2

(Remarks to the Author)

I have reviewed the revised version of the manuscript and appreciate the authors' efforts to address the comments and improve the clarity and quality of the work. I am satisfied with the revisions and support the publication of the manuscript in its current form.

Point-by-point response to reviewers

COMMSBIO-25-0445-T

Bourgade et al.

As R1 and partly R2 pointed out, a better discussion and comparison with other methods of genetic modification of *Synechocystis* must be included.

We agree with R1 that a thorough comparison with production titers of past genetic modifications of *Synechocystis* must be included as well.

Of the specific comments from R2 we would like to highlight that

- 1) negative controls/results must be described better
- 2) critically discussing certain results such as the production levels at later time points or negative impact of CRISPR in X51 background and
- 3) gene expression levels.

We have revised the manuscript to improve clarity and reproducibility. Specifically, we have detailed the negative controls and clarified the results. Additionally, we have extended the discussion, which now includes hypotheses discussing some of the phenomena we observed (e.g. production levels at later time points). Please see below for a point-by-point response to reviewers' comments.

Please avoid overselling in general and specifically mentioning that you present the first example of CRISPRa in *Synechocystis*. CRISPRa was already published, please mention and cite this and if possible, discuss briefly (as far as language translation tools can translate unambiguously) <https://synbioj.cip.com.cn/EN/10.12211/2096-8280.2022-077>.

We have revised the manuscript to include the suggested paper (Lines 70-78 and Lines 366-368) as follows:

*Lines 70-78: "However, in contrast to CRISPRi, CRISPRa remains largely unexplored in cyanobacteria, despite its clear potential to facilitate rapid metabolic mapping and target discovery in both fundamental and applied research. A recent study¹⁹ demonstrated targeted gene activation in *Synechococcus elongatus* PCC 7942 by fusing the native RNA polymerase ω subunit (RpoZ) to dCas9, which was subsequently utilised to enhance isopentenol production. This highlights the potential of CRISPRa as a powerful tool for metabolic engineering in cyanobacteria, offering new opportunities to fine-tune gene expression and optimise biosynthetic pathways."*

*Lines 366-368: "Similar activation levels were observed in *S. elongatus* PCC 7942 using a dCas9-based system, with fold-changes ranging from 2 to 10^{19} , indicating that a limited dynamic range may be an inherent feature of cyanobacterial transcriptional regulation."*

Reviewers' comments:

Reviewer #1 (Remarks to the Author):

The manuscript presents the development and application of a CRISPR activation (CRISPRa) system for the cyanobacterium *Synechocystis* sp. PCC 6803. This system enables robust, multiplexed gene activation of both endogenous and heterologous targets. The authors

employed this approach to identify and upregulate genes influencing isobutanol (IB) and 3-methyl-1-butanol (3M1B) production. Their results demonstrate up to a fourfold increase in compound production through individual and multiplexed gene activation, shedding light on complex regulatory interactions governing biosynthesis. The study provides a valuable tool for high-throughput screening, metabolic pathway optimization, and functional genomics, representing a notable advancement in genetic engineering for *Synechocystis*.

Thank you for reviewing our manuscript and suggesting revisions to improve it. Please see below for a detailed response to your suggestions.

Since CRISPRa has been successfully applied in various organisms, it would strengthen the study to discuss both the common challenges encountered when implementing CRISPRa across different systems and the specific challenges unique to *Synechocystis* or cyanobacteria. Addressing these aspects would help clarify whether adaptations were necessary for the system to function effectively in *Synechocystis* and how these insights might inform future applications in related organisms.

Thank you for this insightful comment. We have expanded the discussion to include challenges that we believe are relevant for our study.

First, CRISPR approaches are inherently dependent on the availability of PAM sequences at candidate loci. Although our tool showed increased targeting flexibility, it remains possible that specific endogenous genes cannot be activated due to the lack of PAM sequences within the targeting window. We now discuss this briefly (Lines 349-352) as follows:

“However, despite this increased targeting flexibility, efficient activation of certain endogenous genes may still be constrained by the absence of suitable PAM sequences within the optimal distance from the TSS – an inherent limitation of CRISPR-based approaches⁴³.”

*In terms of challenges specific to *Synechocystis*, we observed a narrow dynamic range compared to other prokaryotes, achieving low activation levels, as discussed from Line 359 onwards. We hypothesise that this phenomenon is caused by the intrinsic narrow range of transcriptional regulation in place in *Synechocystis*. Additionally, cyanobacteria are polyploid organisms with a fluctuating number of chromosomal copies, which may impact activation levels. We have now included this challenge in the discussion (Lines 374-382) as follows:*

*“Additionally, given that *Synechocystis* is highly polyploid⁴⁸, the high number of chromosomal copies may influence activation efficiency, potentially reducing the overall expression fold-increase. Notably, chromosomal copy number is highly dynamic, varying with environmental conditions, growth stages and even amongst individual cells within the same population⁴⁹, suggesting that activation efficiency may fluctuate over time and between cells. Since our study assessed tool efficiency at the population level, we may overlook potential heterogeneity in gene activation. Single-cell approaches could provide deeper insights into these variations.”*

*Finally, in preliminary experiments, we were unable to express dCas12a using a constitutive promoter in *Synechocystis* (data not presented), suggesting potential toxicity. To mitigate this, we employed a rhamnose-inducible promoter, allowing precise control over dCas12a expression and therefore minimising potential growth defects and metabolic burden associated with its continuous expression.*

Additionally, given that cyanobacteria have been used previously to produce IB and 3M1B from CO₂, comparing the productivities achieved in this study with those reported in prior work without CRISPRa would help contextualize its impact on production efficiency.

Thank you for your comment. To better contextualise our findings, we have chosen to emphasise the fold-change in production in activated strains relative to their respective controls. We believe this approach more clearly highlights the influence of specific metabolic pathways on production levels, providing new research avenues for future projects to increase IB/3M1B production. Notably, the targets explored in this study had not been previously evaluated for their impact on IB/3M1B production, further underscoring the novelty of these findings.

*Additionally, we acknowledge that comparing our findings to previous studies can help contextualise the impacts we report on production. However, we believe direct comparisons may not be appropriate due to significant differences in cultivation setups and conditions across laboratories. In particular, light conditions significantly influence photosynthesis, a key metabolic process affecting production. For instance, a recent study on IB/3M1B production in *Synechocystis* (Kobayashi et al., 2022; doi: 10.1186/s12934-021-01732-x) employed red and/or green light, whereas our experiments were conducted under white light. These variations complicate direct comparisons between studies. While we could compare our results to previous studies from our own laboratory, doing so exclusively would not provide a fair and comprehensive perspective, as it would omit findings from other research groups.*

Additionally, we report production data normalised to cell density to account for growth-related effects, providing a clearer representation of regulatory and metabolic influences at the single-cell level, which fits better with the goal of our study. However, other studies often report titres in mg L⁻¹, further complicating cross-study comparisons.

To clarify this, we have added a short section in the Discussion as follows (Lines 566-570):

“While direct comparisons of production titres across studies remain challenging due to variations in experimental conditions⁶⁴, such as light intensity and culture setup, our approach achieved up to a 4-fold increase in product formation with single-gene activation and up to a 6-fold increase with multiplexed targeting. These results highlight key metabolic pathways limiting IB/3M1B biosynthesis.”

Reviewer #2 (Remarks to the Author):

1. Brief summary of the manuscript

This manuscript presents a CRISPR activation (CRISPRa) system for *Synechocystis* sp. PCC 6803, enabling targeted gene upregulation for metabolic engineering. The authors demonstrate its effectiveness by enhancing biofuel production, showing that activating genes like *pyk1* increases isobutanol and 3-methyl-1-butanol yields. Interestingly, gene activation did not always correlate with metabolite production, suggesting complex regulatory effects. The study provides a useful tool for strain optimization.

2. Overall impression of the work

This study represents a valuable contribution to the genetic toolkit available for cyanobacterial engineering, offering a CRISPRa-based approach for targeted gene activation. The system has potential applications not only in metabolic engineering, such as high-throughput screening and pathway optimization, but also in fundamental studies aimed at understanding the regulation of metabolic pathways in *Synechocystis*. The authors provide a thorough characterization of the CRISPRa system developed, demonstrating the advantages of using dCas12a for precise gene upregulation.

The work is scientifically sound, with well-designed experiments and clear methodology. While further validation under diverse conditions and comparisons with existing approaches would

strengthen its impact, the study presents a promising and useful tool for the field. Overall, the manuscript is of high quality and provides meaningful insights. I support its publication, pending minor revisions to improve clarity and address the points raised.

Thank you for your insightful comments. We have revised the manuscript according to your suggestions to improve clarity and reproducibility. Please see below for a point-by-point response.

3. Specific comments, with recommendations for addressing each comment

Lines 46–48: The statement regarding the limited availability of genetic tools in *Synechocystis* should be nuanced. While improvements are still needed, *Synechocystis* is one of the most extensively studied cyanobacteria, with a well-established genetic toolkit. Consider rephrasing to reflect this more accurately.

We have revised this statement (Lines 44-48) to be more nuanced as follows:

*“In particular, while *Synechocystis* benefits from a more developed genetic toolkit compared to other cyanobacteria, the available tools still lack the diversity and versatility seen in model bacteria, restricting complex metabolic engineering and limiting efforts to optimise compound production.”*

Lines 113–116: The authors report that gRNAs positioned between -97 and -156 yielded the highest fluorescence levels, suggesting this region as optimal for gene activation via dCas12a-SoxS in *Synechocystis*. Could the authors propose a mechanistic explanation for why this region is particularly effective? For instance, could it be related to RNA polymerase accessibility or the presence of specific regulatory elements?

*Thank you for this insightful comment. We propose that the distance between SoxS and RNA polymerase is the key parameter impacting activation levels in our experiments. It has previously been shown that SoxS and RNA polymerase must directly interact for gene upregulation (Dong et al., 2018; Fontana et al., 2020). As SoxS is tethered to dCas12a in our system, we hypothesise that its interaction with RNA polymerase is thus dependent on the distance between the dCas12a binding site and the promoter sequence. This is further supported by a previous study (Fontana et al., 2020) which highlighted that expression at a SoxS-dependent promoter was dependent on the distance between the TSS and SoxS binding site in its native context (*E. coli*).*

We have added the following section in the Discussion section (Lines 339-345) to include this hypothesis in the revised manuscript:

“We hypothesise that this window is determined by the spatial constraints required for direct interactions between the dCas12a-tethered SoxS and RNA polymerase, which are required for successful activation¹¹. As a previous study¹⁹ showed that SoxS-dependent gene expression was highly sensitive to the distance between the TSS and the SoxS binding site in its native context, the positioning of the dCas12a binding site relative to the TSS, dictating the positioning of SoxS, likely determines activation efficiency.”

Figure 2: The description of the negative control should be expanded. Please clarify its exact role in the experiment and how it validates the observed effects.

We have expanded the figure legend to clarify the negative control used in these experiments.

Lines 177–178: "At later time points, IB and 3M1B levels became comparable between CRISPRa-targeted and non-targeted strains." Could the authors provide a hypothesis for this

convergence in production levels? Potential explanations regarding metabolic feedback, resource allocation, or regulatory mechanisms would be valuable.

*While our results do not allow us to confidently determine the cause of this phenomenon, we suggest that future work incorporate multi-omics analyses of different strains across multiple timepoints to understand underlying regulatory and metabolic mechanisms. Nonetheless, we hypothesise that IB/3M1B formation is tightly interconnected with BCAA biosynthesis, which may be regulated with negative feedback loops and allosteric control, as described in other organisms. Notably, *kivD* expression has been shown to respond to BCAA concentrations in *L. lactis*, further suggesting a strong cross-talk between BCAA biosynthesis and IB/3M1B formation. Additionally, KivD enzymatic activity is influenced by pH and ion concentrations, which may fluctuate intracellularly between growth phases, therefore impacting IB/3M1B formation. Finally, *Synechocystis* undergoes significant transcriptional changes in different growth stages, which may reduce the effectiveness of our CRISPRa system as cultures progress into later growth stages.*

We have revised the discussion section (Lines 397-421) to incorporate these additional considerations as follows:

“The native regulation of the branched-chain amino acid (BCAA) biosynthesis in *Synechocystis* remains largely unexplored but directly influences IB/3M1B formation. In other organisms, tight transcriptional and enzymatic control, with negative feedback loops and allosteric inhibition, respectively, has been shown to limit intermediate formation in this metabolic pathway⁵⁰, which may explain the modest improvements, observed in our experiments. Notably, an early study reported that BCAA concentrations influence *kivD* expression and potentially its enzymatic activity in *Lactococcus lactis*⁵¹, further highlighting the intricate relationship between BCAA biosynthesis and IB/3M1B formation. We, thus, suggest that future metabolic engineering efforts should focus on addressing these potential regulatory constraints to substantially enhance cyanobacterial IB/3M1B biosynthesis. Furthermore, our findings suggest that *kivD*^{S286T} upregulation was primarily beneficial for IB/3M1B production at early timepoints, while its impact diminished at later growth stages. This pattern indicates the presence of unknown regulatory and metabolic constraints that limit IB/3M1B biosynthesis. Future research should incorporate multi-omics analyses across different timepoints to identify such underlying mechanisms. KivD activity has been shown to depend on pH and ion concentrations⁵², raising the possibility that intracellular conditions become less favourable for IB/3M1B formation as the culture progresses. Notably, *Synechocystis* cultures have been observed to rapidly alkalinize⁵³ while KivD activity significantly declines at pH levels above 7.5. Moreover, drastic transcriptome-wide changes occur across different growth phases in *Synechocystis*⁵⁴, which may reduce the effectiveness of our CRISPRa system at later stages. Controlling this transcriptional rewiring to maintain strong activation may be challenging, as it likely involves complex regulatory networks that dynamically adapt to cellular and environmental conditions.”

Figure 4: The figure legend should be revised to provide a more complete and clearer explanation of the data presented. Currently, it lacks sufficient detail to fully interpret the results. Specifically, the labels for the different experimental conditions need to be clarified, and the relationship between the data in the figure and the corresponding experimental setup should be explicitly defined. This will improve the readability and ensure that the figure is easily understood by readers.

We have expanded the figure legend of Fig. 4 to improve clarity.

Lines 190–192: The manuscript describes a modest reduction in IB and 3M1B production in some CRISPRa-targeted strains, particularly in HX51 derivatives. Can the authors suggest a possible explanation for this decrease? Could it be linked to metabolic burden, competition for cellular resources, or unintended regulatory effects?

*Thank you for your comment. It is difficult to understand the production reduction observed in some strains with the data we present. Additional experiments, such as omics approaches, would be informative to understand the underlying mechanism. Nonetheless, we hypothesise that genetic and phenotypic instability, which have been identified as key challenges for cyanobacterial biotechnology, play a role in this phenomenon. In particular, considering that HX51 carried three identical copies of the *kivD* construct, potential intra- and inter-chromosomal recombination may have re-organised the genetic landscape, therefore impacting *dCas12a* binding. Additional mechanisms, such as high metabolic burden caused by antibiotic resistance, elevated IB/3M1B formation and high *KivD* expression, may further increase strain instability.*

We have expanded the discussion (Lines 434-445) to include these points in the revised manuscript as follows:

*“Such genetic or phenotypic instability may contribute to the slight reduction of IB/3M1B formation observed in some strains, particularly prevalent in the HX51 background. Given that HX51 strains harbour three copies of an identical *kivD*^{S286T} construct, intra- and inter-chromosomal recombination – an underexplored phenomenon in cyanobacteria⁵⁶ – could potentially rearrange heterologous genetic elements, affecting CRISPRa-mediated gene upregulation. Additionally, metabolic burden from multiple antibiotic resistance, increased carbon flux towards IB/3M1B formation and high proteome allocation to *KivD*, may further exacerbate strain instability, contributing to the observed production decline. Further investigation is required to elucidate the mechanisms underlying the reduced production in some HX51 activated strains, particularly the potential interplay between genetic instability and metabolic constraints.”*

Lines 197–199: The lower activation levels and delayed growth observed in HX11-derived strains (Supplementary Fig. 3) are attributed to an unidentified phenotype in the HX11 background. Could the authors provide further details or specific examples of potential factors influencing this response?

Thank you for this comment. We consistently observed delayed growth in HX11-derived strains across all of our experiments. Since target mapping and multiplexing (Results sections “CRISPRa-mediated metabolic mapping to identify candidate genes enhancing IB and 3M1B biosynthesis” and “Combinatorial effects of CRISPRa/i multiplexing on IB/3M1B biosynthesis”) were also conducted in the HX11 background, we can exclude the possibility that the CRISPRa system is not functional in these strains. However, during our experiments, we were unable to determine the underlying cause for this phenotype. We speculate that IB/3M1B biosynthesis causes an elevated metabolic burden in this background, potentially due to unidentified point mutations throughout the genome or altered regulatory mechanisms. To identify the factors contributing to this growth delay, comprehensive multi-omics analyses would be particularly valuable in uncovering causal mechanisms and providing deeper insights into the observed phenotype.

Lines 187–188: The phrase “dual targeting of the NS1 and *ddh* loci resulted in opposite effects on IB and 3M1B production in HX11” is unclear. If the observed effect was a decrease in production, Figure 4B does not seem to reflect this. Please clarify the intended interpretation.

We applied the same multiplexing strategy in the two background strains HX11 and HX51 to co-target the NS1 and ddh loci. However, we observed opposite effects on production in the resulting strains. In HX11, CRISPRa targeting at NS1 and ddh increased IB and 3M1B titres, as illustrated on Fig. 4B. In contrast, in HX51, CRISPRa targeting at NS1 and ddh unexpectedly led to a reduction of IB and 3M1B biosynthesis, as illustrated on Fig. 4C. We have revised the original sentence as follows to improve clarity (Lines 198-205):

*“Additionally, gRNA performance varied depending on the strain background, as demonstrated by the opposite effects of dual NS1 and ddh targeting in HX11 and HX51. In HX11, co-targeting these two loci led to a modest increase of IB and 3M1B production whereas, in contrast, the same approach in HX51 unexpectedly reduced IB and 3M1B titres. Since identical *kivD*^{S286T} constructs and gRNAs were used for gene activation, this observation indicates that genetic context of the respective strains rather than gRNA efficiency may account for these differences.”*

Lines 189–190: If gRNAs were designed upstream of the promoter, please specify the exact position for each gene and its respective genetic background to facilitate reproducibility.

We have added the following sentence (Lines 182-185) to specify the position for each gRNA:

*“Based on our previous findings identifying an optimal targeting window for CRISPRa, we designed gRNAs binding within this window for each target locus (*ddh*: -115; NS1: -156; *sll1564*: -173) to maximise gene activation.”*

Lines 384–386: "All candidates demonstrated robust activation, validating the CRISPRa system's applicability for precise upregulation of endogenous genes." If referring to Supplementary Figure 6 (qPCR results), a fold-change of 1 does not indicate an actual change in gene expression. This statement should be nuanced accordingly. Additionally, it is important to differentiate between fold-change in IB and 3M1B production and direct upregulation of target genes. A more appropriate approach might be to compare expression levels with a non-activated control.

*Thank you for this comment. We have nuanced the statement (Lines 448-450) as follows to clarify that one of the gRNAs (*pyk2.1*, Suppl. Fig. 7) did not lead to upregulation:*

*“With the exception of gRNA *pyk2.1*, which did not result in successful activation (Supplementary Figure 7), all candidate genes were significantly upregulated, validating the CRISPRa system's applicability for precise upregulation of endogenous genes.”*

*The lack of activation with *pyk2.1* may be due to intrinsic factors, such as binding efficiency or target site accessibility. However, successful upregulation of *pyk2* was achieved using gRNA *pyk2.2*, confirming that all candidate genes (albeit not with all gRNAs) in this experiment were effectively activated (all fold-changes superior to 1.5 – apart from *pyk2.1*).*

As you pointed out in your comment, no clear correlation was observed between fold-increase in IB/3M1B production and gene upregulation, as we discuss from Line 407 onwards. Therefore, both parameters are required when characterising CRISPRa and strain performance.

We believe that comparing activated strains to an activated control lacking a targeting gRNA (EV) subjected to an identical treatment (i.e. rhamnase induction) provides a more appropriate reference. We cannot exclude that rhamnase induction or expression of dCas12a-SoxS has potential regulatory or metabolic impacts, which may affect results when comparing activated VS non-activated samples.

General Considerations:

Negative controls: The description of negative controls should be more detailed for each experiment to improve clarity and ensure reproducibility.

Thank you for this suggestion. We have added a brief description of the negative controls in the relevant figure legends. To further improve clarity, we have added a short paragraph in the Methods section (Lines 615-619) describing the negative controls in more details as follows:

“In all experiments, the negative control, referred to as EV, corresponds to the respective background strain carrying plasmid pBB_CA, which encodes the dCas12a-SoxS fusion and a CRISPR array without a targeting gRNA, both regulated by the rhamnose-inducible system. Unless otherwise stated, EV controls were subjected to the same treatment as the CRISPRa-targeted strains.”

Vector design: Adding a schematic representation of the vector containing multiple gRNAs would enhance the understanding of the experimental setup.

Thank you for this suggestion. We agree that a figure on vector design would improve clarity. We have thus added Supplementary Figure 1, which summarises the vector organisation and schematically illustrates the gRNA structure.

Reviewer #3 (Remarks to the Author):

The manuscript outlines in detail the development of a CRISPR activation system for targeted gene of certain in *Synechocystis* sp. PCC 6803. Overall the manuscript is well presented without any obvious major issues.

Thank you for reviewing our manuscript.